# A SCORE-BASED DENSITY FORMULA, WITH APPLICATIONS IN DIFFUSION GENERATIVE MODELS

## ABSTRACT

Score-based generative models (SGMs) have revolutionized the field of generative modeling, achieving unprecedented success in generating realistic and diverse content. Despite empirical advances, the theoretical basis for why optimizing the evidence lower bound (ELBO) on the log-likelihood is effective for training diffusion generative models, such as DDPMs, remains largely unexplored. In this paper, we address this question by establishing a density formula for a continuous-time diffusion process, which can be viewed as the continuous-time limit of the forward process in an SGM. This formula reveals the connection between the target density and the score function associated with each step of the forward process. Building on this, we demonstrate that the minimizer of the optimization objective for training DDPMs nearly coincides with that of the true objective, providing a theoretical foundation for optimizing DDPMs using the ELBO. Furthermore, we offer new insights into the role of score-matching regularization in training GANs, the use of ELBO in diffusion classifiers, and the recently proposed diffusion loss.

## 1 INTRODUCTION

Score-based generative models (SGMs) represent a groundbreaking advancement in the realm of generative models, significantly impacting machine learning and artificial intelligence by their ability to synthesize high-fidelity data instances, including images, audio, and text (Sohl-Dickstein et al., 2015; Ho et al., 2020; Song et al., 2021b; Song & Ermon, 2019; Dhariwal & Nichol, 2021; Song et al., 2021a). These models operate by progressively refining noisy data into samples that resemble the target distribution. Due to their innovative approach, SGMs have achieved unprecedented success, setting new standards in generative AI and demonstrating extraordinary proficiency in generating realistic and diverse content across various domains, from image synthesis and super-resolution to audio generation and molecular design (Ramesh et al., 2022; Rombach et al., 2022; Saharia et al., 2022; Croitoru et al., 2023; Yang et al., 2023).

The foundation of SGMs is rooted in the principles of stochastic processes, especially stochastic differential equations (SDEs). These models utilize a forward process, which involves the gradual corruption of an initial data sample with Gaussian noise over several time steps. This forward process can be described as:

$$X_0 \overset{\text{add noise}}{\longrightarrow} X_1 \overset{\text{add noise}}{\longrightarrow} \cdots \overset{\text{add noise}}{\longrightarrow} X_T, \tag{1.1}$$

where $X_0 \sim p_{\text{data}}$ is the original data sample, and $X_T$ is a sample close to pure Gaussian noise. The ingenuity of SGMs lies in constructing a reverse denoising process that iteratively removes the noise, thereby reconstructing the data distribution. This reverse process starts from a Gaussian sample $Y_T$ and moves backward as:

$$Y_T \overset{\text{denoise}}{\longrightarrow} Y_{T-1} \overset{\text{denoise}}{\longrightarrow} \cdots \overset{\text{denoise}}{\longrightarrow} Y_0 \tag{1.2}$$

ensuring that $Y_t \overset{\text{d}}{\approx} X_t$ at each step $t$. The final output $Y_0$ is a new sample that closely mimics the distribution of the initial data $p_{\text{data}}$.

Inspired by the classical results on time-reversal of SDEs (Anderson, 1982; Haussmann & Pardoux, 1986), SGMs construct the reverse process guided by score functions $\nabla \log p_{X_t}$ associated with each

step of the forward process. Although these score functions are unknown, they are approximated by neural networks trained through score-matching techniques (Hyvärinen, 2005; 2007; Vincent, 2011; Song & Ermon, 2019). This leads to two popular models: denoising diffusion probabilistic models (DDPMs) (Ho et al., 2020; Nichol & Dhariwal, 2021) and denoising diffusion implicit models (DDIMs) (Song et al., 2021a). While the theoretical results in this paper do not depend on the specific construction of the reverse process, we will use the DDPM framework to discuss their implications for diffusion generative models.

However, despite empirical advances, there remains a lack of theoretical understanding for diffusion generative models. For instance, the optimization target of DDPM is derived from a variational lower bound on the log-likelihood (Ho et al., 2020), which is also referred to as the evidence lower bound (ELBO) (Luo, 2022). It is not yet clear, from a theoretical standpoint, why optimizing a lower bound of the true objective is still a valid approach. More surprisingly, recent research suggests incorporating the ELBO of a pre-trained DDPM into other generative or learning frameworks to leverage the strengths of multiple architectures, effectively using it as a proxy for the negative log-likelihood of the data distribution. This approach has shown empirical success in areas such as GAN training, classification, and inverse problems (Xia et al., 2023; Li et al., 2023a; Graikos et al., 2022; Mardani et al., 2024). While it is conceivable that the ELBO is a reasonable optimization target for training DDPMs (as similar idea is utilized in e.g., the majorize-minimization algorithm), it is more mysterious why it serves as a good proxy for the negative log-likelihood in these applications.

In this paper, we take a step towards addressing the aforementioned question. On the theoretical side, we establish a density formula for a diffusion process $(X_t)_{0 \le t < 1}$ defined by the following SDE:

$$\mathrm{d}X_t = -\frac{1}{2(1-t)}X_t \mathrm{d}t + \frac{1}{\sqrt{1-t}}\mathrm{d}B_t \quad (0 \le t < 1), \qquad X_0 \sim p_{\mathsf{data}},$$

which can be viewed as a continuous-time limit of the forward process (1.1). Under some regularity conditions, this formula expresses the density of $X_0$ with the score function along this process, having the form

$$\log p_{X_0}(x) = -\frac{1+\log(2\pi)}{2}d - \int_0^1 \left[ \frac{1}{2(1-t)}\mathbb{E}\left[ \left\| \frac{X_t - \sqrt{1-t}X_0}{t} + \nabla \log p_{X_t}(X_t) \right\|_2^2 \mid X_0 = x \right] - \frac{d}{2t} \right] \mathrm{d}t,$$

where $p_{X_t}(\cdot)$ is the density of $X_t$. By time-discretization, this reveals the connection between the target density $p_{\mathsf{data}}$ and the score function associated with each step of the forward process (1.1). These theoretical results will be presented in Section 3.

Finally, using this density formula, we demonstrate that the minimizer of the optimization target for training DDPMs (derived from the ELBO) also nearly minimizes the true target—the KL divergence between the target distribution and the generator distribution. This finding provides a theoretical foundation for optimizing DDPMs using the ELBO. Additionally, we use this formula to offer new insights into the role of score-matching regularization in training GANs (Xia et al., 2023), the use of ELBO in diffusion classifiers (Li et al., 2023a), and the recently proposed diffusion loss (Li et al., 2024). These implications will be discussed in Section 4.

## 2 PROBLEM SET-UP

In this section, we formally introduce the Denoising Diffusion Probabilistic Model (DDPM) and the stochastic differential equation (SDE) that describes the continuous-time limit of the forward process of DDPM.

### 2.1 DENOISING DIFFUSION PROBABILISTIC MODEL

Consider the following forward Markov process in discrete time:

$$X_t = \sqrt{1-\beta_t}X_{t-1} + \sqrt{\beta_t}W_t \quad (t = 1, \ldots, T), \qquad X_0 \sim p_{\mathsf{data}}, \tag{2.1}$$

where $W_1, \ldots, W_T \overset{\text{i.i.d.}}{\sim} \mathcal{N}(0, I_d)$ and the learning rates $\beta_t \in (0, 1)$. Since our main results do not depend on the specific choice of $\beta_t$, we will specify them as needed in later discussions. For each

$t \in [T]$, let $q_t$ be the law or density function of $X_t$, and let $\alpha_t := 1 - \beta_t$ and $\overline{\alpha}_t := \prod_{i=1}^{t} \alpha_i$. A simple calculation shows that:

$$X_t = \sqrt{\overline{\alpha}_t} X_0 + \sqrt{1 - \overline{\alpha}_t}\, \overline{W}_t \qquad \text{where} \qquad \overline{W}_t \sim \mathcal{N}(0, I_d). \tag{2.2}$$

We will choose the learning rates $\beta_t$ to ensure that $\overline{\alpha}_T$ is sufficiently small, such that $q_T$ is close to the standard Gaussian distribution.

The key components for constructing the reverse process in the context of DDPM are the score functions $s_t^\star : \mathbb{R}^d \to \mathbb{R}^d$ associated with each $q_t$, defined as the gradient of their log density:

$$s_t^\star(x) := \nabla \log q_t(x) \quad (t = 1, \dots, T).$$

While these score functions are not explicitly known, in practice, noise-prediction networks $\varepsilon_t(x)$ are trained to predict

$$\varepsilon_t^\star(x) := -\sqrt{1 - \overline{\alpha}_t}\, s_t^\star(x),$$

which are often referred to as epsilon predictors. To construct the reverse process, we use:

$$Y_{t-1} = \frac{1}{\sqrt{\alpha_t}} \big( Y_t + \eta_t s_t(Y_t) + \sigma_t Z_t \big) \quad (t = T, \dots, 1), \qquad Y_T \sim \mathcal{N}(0, I_d) \tag{2.3}$$

where $Z_1, \dots, Z_T \overset{\text{i.i.d.}}{\sim} \mathcal{N}(0, I_d)$, and $s_t(\cdot) := -\varepsilon_t(\cdot)/\sqrt{1 - \overline{\alpha}_t}$ is the estimate of the score function $s_t^\star(\cdot)$. Here $\eta_t, \sigma_t > 0$ are the coefficients that influence the performance of the DDPM sampler, and we will specify them as needed in later discussion. For each $t \in [T]$, we use $p_t$ to denote the law or density of $Y_t$.

## 2.2 A CONTINUOUS-TIME SDE FOR THE FORWARD PROCESS

In this paper, we build our theoretical results on the continuous-time limit of the aforementioned forward process, described by the diffusion process:

$$\mathrm{d}X_t = -\frac{1}{2(1-t)} X_t \mathrm{d}t + \frac{1}{\sqrt{1-t}} \mathrm{d}B_t \quad (0 \le t < 1), \qquad X_0 \sim p_{\text{data}}, \tag{2.4}$$

where $(B_t)_{t \ge 0}$ is a standard Brownian motion. The solution to this stochastic differential equation (SDE) has the closed-form expression:

$$X_t = \sqrt{1-t} X_0 + \sqrt{t}\, \overline{Z}_t \qquad \text{where} \qquad \overline{Z}_t = \sqrt{\frac{1-t}{t}} \int_0^t \frac{1}{1-s} \mathrm{d}B_s \sim \mathcal{N}(0, I_d). \tag{2.5}$$

It is important to note that the process $X_t$ is not defined at $t = 1$, although it is straightforward to see from the above equation that $X_t$ converges to a Gaussian variable as $t \to 1$.

To demonstrate the connection between this diffusion process and the forward process (2.1) of the diffusion model, we evaluate the diffusion process at times $t_i = \sqrt{1 - \overline{\alpha}_i}$ for $1 \le i \le T$. It is straightforward to check that the marginal distribution of the resulting discrete-time process $\{X_{t_i} : 1 \le i \le T\}$ is identical to that of the forward process (2.1). Therefore the diffusion process (2.4) can be viewed as a continuous-time limit of the forward process. In the next section, we will establish theoretical results for the diffusion process (2.4). Through time discretization, our theory will provide insights for the DDPM.

We use the notation $X_t$ for both the discrete-time process $\{X_t : t \in [T]\}$ in (2.1) and the continuous-time diffusion process $(X_t)_{0 \le t < 1}$ in (2.4) to maintain consistency with standard literature. The context will clarify which process is being referred to.

## 3 THE SCORE-BASED DENSITY FORMULA

### 3.1 MAIN RESULTS

Our main results are based on the continuous-time diffusion process $(X_t)_{0 \le t < 1}$ defined in (2.4). While $X_0$ might not have a density, for any $t \in (0, 1)$, the random variable $X_t$ has a smooth density, denoted by $\rho_t(\cdot)$. Our main result characterizes the evolution of the conditional mean of $\log \rho_t(X_t)$ given $X_0$, as stated below.

**Theorem 1.** *Consider the diffusion process $(X_t)_{0 \le t < 1}$ defined in (2.4), and let $\rho_t$ be the density of $X_t$. For any $0 < t_1 < t_2 < 1$, we have*

$$\mathbb{E}\left[\log \rho_{t_2}(X_{t_2}) - \log \rho_{t_1}(X_{t_1}) \mid X_0\right] = \int_{t_1}^{t_2} \left(\frac{1}{2(1-t)}\mathbb{E}\left[\left\|\frac{X_t - \sqrt{1-t}X_0}{t} + \nabla \log \rho_t(X_t)\right\|_2^2 \mid X_0\right] - \frac{d}{2t}\right)\mathrm{d}t.$$

The proof of this theorem is deferred to Appendix A. A few remarks are as follows. First, it is worth mentioning that this formula does not describe the evolution of the (conditional) differential entropy of the process, because $\rho_t(\cdot)$ represents the unconditional density of $X_t$, while the expectation is taken conditional on $X_0$. Second, without further assumptions, we cannot set $t_1 = 0$ or $t_2 = 1$ because $X_0$ might not have a density (hence $\rho_0$ is not well-defined), and $X_t$ is only defined for $t < 1$. By assuming that $X_0$ has a finite second moment, the following proposition characterizes the limit of $\mathbb{E}[\log \rho_t(X_t) \mid X_0]$ as $t \to 1$.

**Proposition 1.** *Suppose that $\mathbb{E}[\|X_0\|_2^2] < \infty$. Then for any $x_0 \in \mathbb{R}^d$, we have*

$$\lim_{t \to 1-} \mathbb{E}\left[\log \rho_t(X_t) \mid X_0 = x_0\right] = -\frac{1 + \log(2\pi)}{2}d.$$

The proof of this proposition is deferred to Appendix B. This result is not surprising, as it can be seen from (2.5) that $X_t$ converges to a standard Gaussian variable as $t \to 1$ regardless of $x_0$, and we can check

$$\mathbb{E}[\log \phi(Z)] = -\frac{1 + \log(2\pi)}{2}d$$

where $Z \sim \mathcal{N}(0, I_d)$ and $\phi(\cdot)$ is its density (we will use this notation throughout his section). The proof of Proposition 1 formalizes this intuitive analysis.

When $X_0$ has a smooth density $\rho_0(\cdot)$ with Lipschitz continuous score function, we can show that $\mathbb{E}[\log \rho_t(X_t) \mid X_0] \to \rho_0(x_0)$ as $t \to 0$, as presented in the next proposition.

**Proposition 2.** *Suppose that $X_0$ has density $\rho_0(\cdot)$ and $\sup_x \|\nabla^2 \log \rho_0(x)\| < \infty$. Then for any $x_0 \in \mathbb{R}^d$, we have*

$$\lim_{t \to 0+} \mathbb{E}\left[\log \rho_t(X_t) \mid X_0 = x_0\right] = \log \rho_0(x_0).$$

The proof of this proposition can be found in Appendix C. With Propositions 1 and 2 in place, we can take $t_1 \to 0$ and $t_2 \to 1$ in Theorem 1 to show that for any given point $x_0$,

$$\log \rho_0(x_0) = -\frac{1 + \log(2\pi)}{2}d - \int_0^1 D(t, x_0)\mathrm{d}t \tag{3.1a}$$

where the function $D(x, t)$ is defined as

$$D(t, x) := \frac{1}{2(1-t)}\mathbb{E}\left[\left\|\frac{X_t - \sqrt{1-t}X_0}{t} + \nabla \log \rho_t(X_t)\right\|_2^2 \mid X_0 = x\right] - \frac{d}{2t}. \tag{3.1b}$$

In practice, we might not want to make smoothness assumptions on $X_0$ as in Proposition 2. In that case, we can fix some sufficiently small $\delta > 0$ and obtain a density formula

$$\mathbb{E}\left[\log \rho_\delta(X_\delta) \mid X_0 = x_0\right] = -\frac{1 + \log(2\pi)}{2}d - \int_\delta^1 D(t, x_0)\mathrm{d}t \tag{3.1c}$$

for a smoothed approximation of $\log \rho_0(x_0)$. This kind of proximity is often used to circumvent non-smoothness target distributions in diffusion model literature (e.g., Li et al. (2023b); Chen et al. (2022; 2023b); Benton et al. (2023)). We leave some more discussions to Appendix D.

## 3.2 FROM CONTINUOUS TIME TO DISCRETE TIME

In this section, to avoid ambiguity, we will use $(X_t^{\mathsf{sde}})_{0 \le t < 1}$ to denote the continuous-time diffusion process (2.4) studied in the previous section, while keep using $\{X_t : 1 \le t \le T\}$ to denote the forward process (2.1). The density formula (3.1) is not readily implementable because of its continuous-time nature. Consider time discretization over the grid

$$0 < t_1 < t_2 < \cdots < t_T < t_{T+1} = 1 \qquad \text{where} \qquad t_i := 1 - \overline{\alpha}_i \quad (1 \le i \le T).$$

Recall that the forward process $X_1, \ldots, X_T$ has the same marginal distribution as $X_{t_1}^{\mathsf{sde}}, \ldots, X_{t_T}^{\mathsf{sde}}$ snapshoted from the diffusion process (2.4). This gives the following approximation of the density formula (3.1a):

$$\log \rho_0(x_0) \overset{\text{(i)}}{\approx} \mathbb{E}\left[\log \rho_{t_1}(X_{t_1}^{\mathsf{sde}}) \,|\, X_0^{\mathsf{sde}} = x_0\right]$$

$$\overset{\text{(ii)}}{\approx} -\frac{1 + \log(2\pi t_1)}{2}d - \sum_{i=1}^{T} \frac{t_{i+1} - t_i}{2(1 - t_i)}\mathbb{E}\left[\left\|\frac{X_{t_i}^{\mathsf{sde}} - \sqrt{1 - t_i}X_0^{\mathsf{sde}}}{t_i} + \nabla \log \rho_{t_i}(X_t^{\mathsf{sde}})\right\|_2^2 \,\Big|\, X_0^{\mathsf{sde}} = x_0\right]$$

$$\overset{\text{(iii)}}{\approx} -\frac{1 + \log(2\pi t_1)}{2}d - \sum_{i=1}^{T} \frac{t_{i+1} - t_i}{2t_i(1 - t_i)}\mathbb{E}_{\varepsilon \sim \mathcal{N}(0, I_d)}\left[\left\|\varepsilon - \widehat{\varepsilon}_i(\sqrt{1 - t_i}x_0 + \sqrt{t_i}\varepsilon)\right\|_2^2\right].$$

In step (i) we approximate $\log \rho_0(x_0)$ with a smoothed proxy; see the discussion around (3.1c) for details; step (ii) applies (3.1c), where we compute the integral $\int_{t_1}^1 d/(2t)\mathrm{d}t = -(d/2)\log t_1$ in closed form and approximate the integral

$$\int_{t_1}^1 \frac{1}{2(1 - t)}\mathbb{E}\left[\left\|\frac{X_t^{\mathsf{sde}} - \sqrt{1 - t}X_0^{\mathsf{sde}}}{t} + \nabla \log \rho_t(X_t^{\mathsf{sde}})\right\|_2^2 \,\Big|\, X_0^{\mathsf{sde}} = x_0\right]\mathrm{d}t;$$

step (iii) follows from $X_{t_i}^{\mathsf{sde}} \overset{\mathrm{d}}{=} \sqrt{1 - t_i}x_0 + \sqrt{t_i}\varepsilon$ for $\varepsilon \sim \mathcal{N}(0, I_d)$ conditional on $X_0^{\mathsf{sde}} = x_0$, and the relation

$$\nabla \log \rho_{t_i} = \nabla \log q_i = s_i^\star(x) = -\sqrt{t_i}\varepsilon_i^\star(x) \approx -\sqrt{t_i}\widehat{\varepsilon}_i(x).$$

In practice, we need to choose the learning rates $\{\beta_t : 1 \le t \le T\}$ such that the grid becomes finer as $T$ becomes large. More specifically, we require

$$t_{i+1} - t_i = \overline{\alpha}_i - \overline{\alpha}_{i+1} = \overline{\alpha_i}\beta_{i+1} \le \beta_{i+1} \quad (1 \le i \le T - 1)$$

to be small (roughly of order $O(1/T)$), and $t_1 = \beta_1$ and $1 - t_T = \overline{\alpha}_T$ to be vanishingly small (of order $T^{-c}$ for some sufficiently large constant $c > 0$); see e.g., Li et al. (2023b); Benton et al. (2023) for learning rate schedules satisfying these properties. Finally, we replace the time steps $\{t_i : 1 \le i \le T\}$ with the learning rates for the forward process to achieve[1]

$$\log \rho_0(x_0) \approx -\frac{1 + \log(2\pi\beta_1)}{2}d - \sum_{t=1}^{T} \frac{1 - \alpha_{t+1}}{2(1 - \overline{\alpha}_t)}\mathbb{E}_{\varepsilon \sim \mathcal{N}(0, I_d)}\left[\left\|\varepsilon - \widehat{\varepsilon}_t(\sqrt{\overline{\alpha}_t}x_0 + \sqrt{1 - \overline{\alpha}_t}\varepsilon)\right\|_2^2\right],$$

$$(3.2)$$

The density approximation (3.2) can be evaluated with the trained epsilon predictors.

### 3.3 COMPARISON WITH OTHER RESULTS

The density formulas (3.1) expresses the density of $X_0$ using the score function along the continuous-time limit of the forward process of the diffusion model. Other forms of score-based density formulas can be derived using normalizing flows. Notice that the probability flow ODE of the SDE (2.4) is

$$\dot{x}_t = v_t(x_t) \qquad \text{where} \qquad v_t(x) = -\frac{x - \nabla \log \rho_t(x)}{2(1 - t)}; \qquad (3.3)$$

namely, if we draw a particle $x_0 \sim \rho_0$ and evolve it according to the ODE (3.3) to get the trajectory $t \to x_t$ for $t \in [0, 1)$, then $x_t \sim \rho_t$. See e.g., Song et al. (2021b, Appendix D.1) for the derivation of this result.

Under some smoothness condition, we can use the results developed in Grathwohl et al. (2019); Albergo et al. (2023) to show that for any given $x_0$

$$\log \rho_t(x_t) - \log \rho_0(x_0) = -\int_0^t \mathsf{Tr}\left(\frac{\partial}{\partial x}v_s(x_s)\right)\mathrm{d}s = \int_0^t \frac{d - \mathsf{tr}(\nabla^2 \log \rho_s(x_s))}{2(1 - s)}\mathrm{d}s. \quad (3.4)$$

Here $t \to x_t$ is the solution to the ODE (3.3) with initial condition $x_0$. Since the ODE system (3.3) is based on the score functions (hence $x_t$ can be numerically solved), and the integral in (3.4) is

---

[1]Here we define $\alpha_{T+1} = 0$ to accommodate the last term in the summation.

based on the Jacobian of the score functions, we may take $t \to 1$ and use the fact that $\rho_t(\cdot) \to \phi(\cdot)$ to obtain a score-based density formula

$$\log \rho_0(x_0) = -\frac{d}{2} \log(2\pi) - \frac{1}{2} \|x_1\|_2^2 - \int_0^1 \frac{d - \operatorname{tr}(\nabla^2 \log \rho_s(x_s))}{2(1-s)} \mathrm{d}s. \tag{3.5}$$

However, numerically, this formula is more difficult to compute than our formula (3.1) for the following reasons. First, (3.5) involves the Jacobian of the score functions, which are more challenging to estimate than the score functions themselves. In fact, existing convergence guarantees for DDPM do not depend on the accurate estimation of the Jacobian of the score functions (Benton et al., 2023; Chen et al., 2023a; 2022; Li & Yan, 2024). Second, using this density formula requires solving the ODE (3.3) accurately to obtain $x_1$, which might not be numerically stable, especially when the score function is not accurately estimated at early stages, due to error propagation. In contrast, computing (3.1) only requires evaluating a few Gaussian integrals (which can be efficiently approximated by the Monte Carlo method) and is more stable to score estimation error.

## 4 IMPLICATIONS

In the previous section, we established a density formula

$$\log q_0(x) \approx \underbrace{-\frac{1 + \log (2\pi\beta_1)}{2} d}_{=:C_0^\star} - \sum_{t=1}^{T} \underbrace{\frac{1 - \alpha_{t+1}}{2(1 - \overline{\alpha}_t)} \mathbb{E}_{\varepsilon \sim \mathcal{N}(0, I_d)} \left[ \left\| \varepsilon - \varepsilon_t^\star(\sqrt{\overline{\alpha}_t} x + \sqrt{1 - \overline{\alpha}_t} \varepsilon) \right\|_2^2 \right]}_{=:L_{t-1}^\star(x)} \tag{4.1}$$

up to discretization error (which vanishes as $T$ becomes large) and score estimation error. In this section, we will discuss the implications of this formula in various generative and learning frameworks.

### 4.1 CERTIFYING THE VALIDITY OF OPTIMIZING ELBO IN DDPM

The seminal work (Ho et al., 2020) established the variational lower bound (VLB), also known as the evidence lower bound (ELBO), of the log-likelihood

$$\log p_0(x) \geq -\sum_{t=2}^{T} \underbrace{\mathbb{E}_{x_t \sim p_{X_t | X_0}(\cdot \,|\, x)} \mathsf{KL} \left( p_{X_{t-1} | X_t, X_0}(\cdot \,|\, x_t, x) \,\|\, p_{Y_{t-1} | Y_t}(\cdot \,|\, x_t) \right)}_{=:L_{t-1}(x)}$$

$$- \underbrace{\mathsf{KL} \left( p_{Y_T}(\cdot) \,\|\, p_{X_T | X_0}(\cdot \,|\, x) \right)}_{=:L_T(x)} + \underbrace{\mathbb{E}_{x_1 \sim p_{X_1 | X_0}(\cdot \,|\, x)} \left[ \log p_{Y_0 | Y_1}(x \,|\, x_1) \right]}_{=:C_0(x)}, \tag{4.2}$$

where the reverse process $(Y_t)_{0 \leq t \leq T}$ was defined in Section 2.1, and $p_0$ is the density of $Y_0$. Under the coefficient design recommended by Li & Yan (2024) (other reasonable designs also lead to similar conclusions)

$$\eta_t = 1 - \alpha_t \qquad \text{and} \qquad \sigma_t^2 = \frac{(1 - \alpha_t)(\alpha_t - \overline{\alpha}_t)}{1 - \overline{\alpha}_t}, \tag{4.3}$$

it can be computed that for each $2 \leq t \leq T$:

$$L_{t-1}(x) = \frac{1 - \alpha_t}{2(\alpha_t - \overline{\alpha}_t)} \mathbb{E}_{\varepsilon \sim \mathcal{N}(0, I_d)} \left[ \left\| \varepsilon - \varepsilon_t(\sqrt{\overline{\alpha}_t} x + \sqrt{1 - \overline{\alpha}_t} \varepsilon) \right\|_2^2 \right].$$

We can verify that (i) for each $2 \leq t \leq T$, the coefficients in $L_{t-1}$ from (4.2) and $L_{t-1}^\star$ from (4.1) are identical up to higher-order error; (ii) when $T$ is large, $L_T$ becomes vanishingly small; and (iii) the function

$$C_0(x) = -\frac{1 + \log (2\pi\beta_1)}{2} d + O(\beta_1) = C_0^\star + O(\beta_1)$$

is nearly a constant. See Appendix E.1 for details. It is worth highlighting that as far as we know, existing literature haven't pointed out that $C_0(x)$ is nearly a constant. For instance, Ho et al. (2020) discretize this term to obtain discrete log-likelihood (see Section 3.3 therein), which is unnecessary

in view of our observation. Additionally, some later works falsely claim that $C_0(x)$ is negligible, as we will discuss in the following sections.

Now we discuss the validity of optimizing the variational bound for training DDPMs. Our discussion shows that

$$\underbrace{\mathsf{KL}(q_0 \parallel p_0)}_{=:\mathcal{L}(\varepsilon_1,\dots,\varepsilon_T)} = -\mathbb{E}_{x \sim q_0}[\log p_0(x)] - H(q_0) \leq \underbrace{\mathbb{E}_{x \sim q_0}[L(x)] - C_0^\star - H(q_0) + o(1)}_{=:\mathcal{L}_{\mathsf{vb}}(\varepsilon_1,\dots,\varepsilon_T)}, \quad (4.4)$$

where $H(q_0) = -\int \log q_0(x)\mathrm{d}q_0$ is the entropy of $q_0$, and $L(x)$ denotes the widely used (negative) ELBO[2]

$$L(x) \coloneqq \sum_{t=1}^{T} \frac{1 - \alpha_{t+1}}{2(1 - \overline{\alpha}_t)} \mathbb{E}_{\varepsilon \sim \mathcal{N}(0, I_d)}\left[\left\|\varepsilon - \varepsilon_t(\sqrt{\overline{\alpha}_t}x + \sqrt{1 - \overline{\alpha}_t}\varepsilon)\right\|_2^2\right].$$

The true objective of DDPM is to learn the epsilon predictors $\varepsilon_1, \dots, \varepsilon_T$ that minimizes $\mathcal{L}$ in (4.4), while in practice, the optimization target is the variational bound $\mathcal{L}_{\mathsf{vb}}$. It is known that the global minimizer for

$$\mathbb{E}_{x \sim q_0}[L(x)] = \sum_{t=1}^{T} \frac{1 - \alpha_{t+1}}{2(1 - \overline{\alpha}_t)} \mathbb{E}_{x \sim q_0, \varepsilon \sim \mathcal{N}(0, I_d)}\left[\left\|\varepsilon - \varepsilon_t(\sqrt{\overline{\alpha}_t}x + \sqrt{1 - \overline{\alpha}_t}\varepsilon)\right\|_2^2\right] \quad (4.5)$$

is exactly $\widehat{\varepsilon}_t(\cdot) \equiv \varepsilon_t^\star(\cdot)$ for each $1 \leq t \leq T$ (see Appendix E.1). Although in practice the optimization is based on samples from the target distribution $q_0$ (instead of the population level expectation over $q_0$) and may not find the exact global minimizer, we consider the ideal scenario where the learned epsilon predictors $\widehat{\varepsilon}_t$ equal $\varepsilon_t^\star$ to facilitate discussion. When $\varepsilon_t = \varepsilon_t^\star$ for each $t$, according to (4.1), we have

$$L(x) \approx -\log q_0(x) + C_0^\star. \quad (4.6)$$

Taking (4.4) and (4.6) together gives

$$0 \leq \mathcal{L}(\widehat{\varepsilon}_1, \dots, \widehat{\varepsilon}_T) \leq \mathcal{L}_{\mathsf{vb}}(\widehat{\varepsilon}_1, \dots, \widehat{\varepsilon}_T) \approx -\mathbb{E}_{x \sim q_0}[\log q(x)] + C_0^\star - C_0^\star - H(q_0) = 0, \quad (4.7)$$

namely the minimizer for $\mathcal{L}_{\mathsf{vb}}$ approximately minimizes $\mathcal{L}$, and the optimal value is asymptotically zero when the number of steps $T$ becomes large. This suggests that by minimizing the variational bound $\mathcal{L}_{\mathsf{vb}}$, the resulting generator distribution $p_0$ is guaranteed to be close to the target distribution $q_0$ in KL divergence.

Some experimental evidence suggests that using reweighted coefficients can marginally improve empirical performance. For example, Ho et al. (2020) suggests that in practice, it might be better to use uniform coefficients in the ELBO

$$L_{\mathsf{simple}}(x) \coloneqq \frac{1}{T} \sum_{i=1}^{T} \mathbb{E}_{\varepsilon \sim \mathcal{N}(0, I_d)}\left[\left\|\varepsilon - \widehat{\varepsilon}_{t_i}(\sqrt{\overline{\alpha}_t}x + \sqrt{1 - \overline{\alpha}_t}\varepsilon)\right\|_2^2\right] \quad (4.8)$$

when trainging DDPM to improve sampling quality.[3] This strategy has been adopted by many later works. In the following sections, we will discuss the role of using the ELBO in different applications. While the original literature might use the modified ELBO (4.8), in our discussion we will stick to the original ELBO (4.6) to gain intuition from our theoretical findings.

## 4.2 UNDERSTANDING THE ROLE OF REGULARIZATION IN GAN

Generative Adversarial Networks (GANs) are a powerful and flexible framework for learning the unknown probability distribution $p_{\mathsf{data}}$ that generates a collection of training data (Goodfellow et al., 2014). GANs operate on a game between a generator $G$ and a discriminator $D$, typically

---

[2]We follow the convention in existing literature to remove the last two terms $L_T(x)$ and $C_0(x)$ from (4.2) in the ELBO.

[3]Note that the optimal epsilon predictors $\widehat{\varepsilon}_t$ for $L$ and $L_{\mathsf{simple}}$ are the same, but in practice, we may not find the optimal predictors. This practical strategy is beyond the scope of our theoretical result, and implies that the influence of terms from different steps needs more careful investigation. We conjecture that this is mainly because the estimation error for terms when $t$ is close to zero is larger, hence smaller coefficients for these terms can improve performance.

implemented using neural networks. The generator $G$ takes a random noise vector $z$ sampled from a simple distribution $p_{\text{noise}}$ (e.g., Gaussian) and maps it to a data sample resembling the training data, aiming for the distribution of $G(z)$ to be close to $p_{\text{data}}$. Meanwhile, the discriminator $D$ determines whether a sample $x$ is real (i.e., drawn from $p_{\text{data}}$) or fake (i.e., produced by the generator), outputting the probability $D(x)$ of the former. The two networks engage in a zero-sum game:

$$\min_G \max_D V(G, D) := \mathbb{E}_{x \sim p_{\text{data}}}[\log D(x)] + \mathbb{E}_{z \sim p_{\text{noise}}}[\log(1 - D(G(z)))],$$

with the generator striving to produce realistic data while the discriminator tries to distinguish real data from fake. The generator and discriminator are trained iteratively[4]

$$D \leftarrow \arg\min -\mathbb{E}_{x \sim p_{\text{data}}}[\log D(x)] - \mathbb{E}_{z \sim p_{\text{noise}}}[\log(1 - D(G(z)))],$$
$$G \leftarrow \arg\min -\mathbb{E}_{z \sim p_{\text{noise}}}[\log D(G(z))]$$

to approach the Nash equilibrium $(G^\star, D^\star)$, where the distribution of $G^\star(z)$ with $z \sim p_{\text{noise}}$ matches the target distribution $p_{\text{data}}$, and $D(x) = 1/2$ for all $x$.

It is believed that adding a regularization term to make the generated samples fit the VLB can improve the sampling quality of the generative model. For example, Xia et al. (2023) proposed adding the VLB $L(x)$ as a regularization term to the objective function, where $\{\widehat{\varepsilon}_{t_i}(\cdot) : 1 \leq i \leq T\}$ are the learned epsilon predictors for $p_{\text{data}}$. The training procedure then becomes

$$D \leftarrow \arg\min -\mathbb{E}_{x \sim p_{\text{data}}}[\log D(x)] - \mathbb{E}_{z \sim p_{\text{noise}}}[\log(1 - D(G(z)))],$$
$$G \leftarrow \arg\min -\mathbb{E}_{z \sim p_{\text{noise}}}[\log D(G(z))] + \lambda \mathbb{E}_{z \sim p_{\text{noise}}}[L(G(z))],$$

where $\lambda > 0$ is some tuning parameter. However, it remains unclear what exactly is optimized through the above objective. According to our theory, $L(x) \approx -\log p_{\text{data}}(x) + C_0^\star$. Assuming that this approximation is exact for intuitive understanding, the unique Nash equilibrium $(G_\lambda, D_\lambda)$ satisfies

$$p_{G_\lambda}(x) = \left(z p_{\text{data}}(x)^\lambda - 1\right)_+ p_{\text{data}}(x)$$

for some normalizing factor $z > 0$, where $p_{G_\lambda}$ is the density of $G_\lambda(z)$ with $z \sim p_{\text{noise}}$. See Appendix E.2 for details. This can be viewed as amplifying the density $p_{\text{data}}$ wherever it is not too small, while zeroing out the density where $p_{\text{data}}$ is vanishingly small (which is difficult to estimated accurately), thus improving the sampling quality.

### 4.3 Confirming the use of ELBO in diffusion classifier

Motivated by applications like image classification and text-to-image diffusion model, we consider a joint underlying distribution $p_0(x, c)$, where typically $x$ is the image data and the latent variable $c$ is the class index or text embedding, taking values in a finite set $\mathcal{C}$. For each $c \in \mathcal{C}$, we train a diffusion model for the conditional data distribution $p_0(x \mid c)$, which provides a set of epsilon predictors $\{\widehat{\varepsilon}_t(x; c) : 1 \leq t \leq T, c \in \mathcal{C}\}$. Assuming a uniform prior over $\mathcal{C}$, we can use Bayes' formula to obtain:

$$p_0(c \mid x) = \frac{p_0(c) \, p_0(x \mid c_i)}{\sum_{j \in \mathcal{C}} p_0(c_j) \, p_0(x \mid c_j)} = \frac{p_0(x \mid c)}{\sum_{j \in \mathcal{C}} p_0(x \mid c_j)}.$$

for each $c \in \mathcal{C}$. Recent work (Li et al., 2023a) proposed to use the ELBO[5]

$$-L(x; c) := -\sum_{t=1}^T \frac{1 - \alpha_{t+1}}{2(1 - \overline{\alpha}_t)} \mathbb{E}_{\varepsilon \sim \mathcal{N}(0, I_d)} \left[ \left\| \varepsilon - \widehat{\varepsilon}_t(\sqrt{\overline{\alpha}_t} x + \sqrt{1 - \overline{\alpha}_t} \varepsilon; c) \right\|_2^2 \right]$$

as an approximate class-conditional log-likelihood $\log p_0(x \mid c)$ for each $c \in \mathcal{C}$, which allows them to obtain a posterior distribution

$$\widehat{p}_0(c \mid x) = \frac{\exp(-L(x; c))}{\sum_{j \in \mathcal{C}} \exp(-L(x; c_j))}. \tag{4.9}$$

---

[4]While the most natural update rule for the generator is $G \leftarrow \arg\min \mathbb{E}_{z \sim p_{\text{noise}}}[\log(1 - D(G(z)))]$, both schemes are used in practice and have similar performance. Our choice is for consistency with Xia et al. (2023), and our analysis can be extended to the other choice.

[5]The original paper adopted uniform coefficients; see the last paragraph of Section 4.1 for discussion.

Our theory suggests that $-L(x;c) \approx \log p_0(x \,|\, c) - C_0^\star$, where $C_0^\star = -[1 + \log(2\pi\beta_1)]d/2$ is a universal constant that does not depend on $p_0$ and $c$. This implies that

$$\widehat{p}_0\,(c \,|\, x) \approx \frac{\exp\left(\log p_0(x \,|\, c) - C_0^\star\right)}{\sum_{j \in \mathcal{C}} \exp\left(\log p_0(x \,|\, c_j) - C_0^\star\right)} = \frac{p_0\,(x \,|\, c)}{\sum_{j \in \mathcal{C}} p_0\,(x \,|\, c_j)} = p_0\,(c \,|\, x)$$

providing theoretical justification for using the computed posterior $\widehat{p}_0$ in classification tasks.

It is worth mentioning that, although this framework was proposed in the literature (Li et al., 2023a), it remains a heuristic method before our work. For example, in general, replacing the intractable log-likelihood with a lower bound does not guarantee good performance, as they might not be close. Additionally, recall that there is a term $C_0(x)$ in the ELBO (4.2). Li et al. (2023a) claimed that "*Since $T = 1000$ is large and $\log p_\theta(x_0 \,|\, x_1, c)$ is typically small, we choose to drop this term*". However this argument is not correct, as we already computed in Section 4.1 that this term

$$C_0(x) = -\frac{1 + \log\left(2\pi\beta_1\right)}{2}d + O(\beta_1)$$

can be very large since $\beta_1$ is typically very close to 0. In view of our results, the reason why this term can be dropped is that it equals a universal constant that does not depend on the image data $x$ and the class index $c$, thus it does not affect the posterior (4.9).

### 4.4 DEMYSTIFYING THE DIFFUSION LOSS IN AUTOREGRESSIVE MODELS

Finally, we use our results to study a class of diffusion loss recently introduced in Li et al. (2024), in the context of autoregressive image generation. Let $x^k$ denote the next token to be predicted, and $z$ be the condition parameterized by an autoregressive network $z = f(x^1, \ldots, x^{k-1})$ based on previous tokens as input. The goal is to train the network $z = f(\cdot)$ together with a diffusion model $\{\varepsilon_t(\cdot; z) : 1 \le t \le T\}$ such that $\widehat{p}(x \,|\, z)$ (induced by the diffusion model) with $z = f(x^1, \ldots, x^{k-1})$ can predict the next token $x^k$.

The diffusion loss is defined as follows: for some weights $w_t \ge 0$, let

$$L(z, x) = \sum_{t=1}^{T} w_t \mathbb{E}_{\varepsilon \sim \mathcal{N}(0, I_d)}\left[\left\|\varepsilon - \varepsilon_t(\sqrt{\overline{\alpha_t}}x + \sqrt{1 - \overline{\alpha_t}}\varepsilon; z)\right\|_2^2\right]. \tag{4.10}$$

With training data $\{(x_i^1, \ldots, x_i^k) : 1 \le i \le n\}$, we can train the autoregressive network $f(\cdot)$ and the diffusion model by minimizing the following empirical risk:

$$\arg\min_{f, \varepsilon_1, \ldots, \varepsilon_T} \frac{1}{n}\sum_{i=1}^{n} L\left(f(x_i^1, \ldots, x_i^{k-1}), x_i^k\right). \tag{4.11}$$

To gain intuition from our theoretical results, we take the weights in the diffusion loss (4.10) to be the coefficients in the ELBO (4.6), and for each $z$, suppose that the learned diffusion model for $p(x^k \,|\, z)$ is already good enough, which returns the set of epsilon predictors $\{\widehat{\varepsilon}_t(\cdot; z) : 1 \le t \le T\}$ for the probability distribution of $x^k$ conditioned on $z$. Under this special case, our approximation result (4.6) shows that

$$L(z, x) \approx -\log p(x \,|\, z) + C_0^\star,$$

which suggests that the training objective for the network $f$ in (4.11) can be viewed as approximate MLE, as the loss function

$$\frac{1}{n}\sum_{i=1}^{n} L\left(f(x_i^1, \ldots, x_i^{k-1}), x_i^k\right) \approx -\frac{1}{n}\sum_{i=1}^{n} \log p(x_i^k \,|\, f(x_i^1, \ldots, x_i^{k-1})) + C_0^\star$$

represents the negative log-likelihood function (up to an additive constant) of the observed $x_1^k, \ldots, x_n^k$ in terms of $f$.

## 5 DISCUSSION

This paper develops a score-based density formula that expresses the density function of a target distribution using the score function along a continuous-time diffusion process that bridges this

distribution and standard Gaussian. By connecting this diffusion process with the forward process of score-based diffusion models, our results provide theoretical support for training DDPMs by optimizing the ELBO, and offer novel insights into several applications of diffusion models, including GAN training and diffusion classifiers.

Our work opens several directions for future research. First, our theoretical results are established for the continuous-time diffusion process. It is crucial to carefully analyze the error induced by time discretization, which could inform the number of steps required for the results in this paper to be valid in practice. Additionally, while our results provide theoretical justification for using the ELBO (4.6) as a proxy for the negative log-likelihood of the target distribution, they do not cover other practical variants of ELBO with modified weights (e.g., the simplified ELBO (4.8)). Extending our analysis to other diffusion processes might yield new density formulas incorporating these modified weights. Lastly, further investigation is needed into other applications of this score-based density formula, including density estimation and inverse problems.

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

# A PROOF OF THEOREM 1

Recall the definition of the stochastic process $(X_t)_{0 \leq t \leq 1}$

$$dX_t = -\frac{1}{2(1-t)}X_t dt + \frac{1}{\sqrt{1-t}}dB_t.$$

Define $Y_t := X_t/\sqrt{1-t}$ for any $0 \leq t < 1$, and let $f(t,x) = x/\sqrt{1-t}$, we can use Itô's formula to show that

$$dY_t = df(t, X_t) = \frac{\partial f}{\partial t}(t, X_t)dt + \nabla_x f(t, X_t)^\top dX_t + \frac{1}{2}dX_t^\top \nabla_x^2 f(t, X_t)dX_t$$

$$= \frac{X_t}{2(1-t)^{3/2}}dt + \frac{1}{\sqrt{1-t}}\left(-\frac{1}{2(1-t)}X_t dt + \frac{1}{\sqrt{1-t}}dB_t\right) = \frac{dB_t}{1-t}. \tag{A.1}$$

Therefore the Itô process $Y_t$ is a martingale, which is easier to handle. Let $g(t,y) = \log \rho_t(\sqrt{1-t}y)$, and we can express $\log \rho_t(x) = g(t, x/\sqrt{1-t})$. In view of Itô's formula, we have

$$d \log \rho_t(X_t) = dg(t, Y_t) \overset{(i)}{=} \frac{\partial g}{\partial t}(t, Y_t)dt + \nabla_y g(t, Y_t)^\top dY_t + \frac{1}{2}dY_t^\top \nabla_y^2 g(t, Y_t)dY_t$$

$$\overset{(ii)}{=} \frac{\partial g}{\partial t}(t, Y_t)dt + \frac{1}{1-t}\nabla_y g(t, Y_t)^\top dB_t + \frac{1}{2(1-t)^2}dB_t^\top \nabla_y^2 g(t, Y_t)dB_t$$

$$\overset{(iii)}{=} \frac{\partial g}{\partial t}(t, Y_t)dt + \frac{1}{1-t}\nabla_y g(t, Y_t)^\top dB_t + \frac{1}{2(1-t)^2}\text{tr}\left(\nabla_y^2 g(t, Y_t)\right)dt. \tag{A.2}$$

Here step (i) follows from the Itô rule, step (ii) utilizes (A.1), while step (iii) can be derived from the Itô calculus. Then we investigate the three terms above. Notice that

$$\nabla_y g(t,y)\,|_{y=Y_t} = \frac{\nabla_y \rho_t(\sqrt{1-t}y)}{\rho_t(\sqrt{1-t}Y_t)}\,|_{y=Y_t} = \frac{\nabla_x \rho_t(X_t)\sqrt{1-t}}{\rho_t(X_t)} = \sqrt{1-t}\nabla \log \rho_t(X_t), \tag{A.3}$$

and similarly, we have

$$\nabla_y^2 g(t,y)\,|_{y=Y_t} = (1-t)\nabla^2 \log \rho_t(X_t). \tag{A.4}$$

Substituting (A.3) and (A.4) back into (A.2) gives

$$d \log \rho_t(X_t) = \frac{\partial g}{\partial t}(t, Y_t)dt + \frac{1}{\sqrt{1-t}}\nabla \log \rho_t(X_t)^\top dB_t + \frac{1}{2(1-t)}\text{tr}\left(\nabla^2 \log \rho_t(X_t)\right)dt.$$

or equivalently, for any given $0 < t_1 < t_2 < 1$, we have

$$\log \rho_t(X_t)\Big|_{t_1}^{t_2} = \int_{t_1}^{t_2}\left[\frac{\partial g}{\partial t}(t, Y_t) + \frac{\text{tr}\left(\nabla^2 \log \rho_t(X_t)\right)}{2(1-t)}\right]dt + \int_{t_1}^{t_2}\frac{1}{\sqrt{1-t}}\nabla \log \rho_t(X_t)^\top dB_t. \tag{A.5}$$

Conditional on $X_0$, we take expectation on both sides of (A.5) to achieve

$$\mathbb{E}\left[\log \rho_{t_2}(X_{t_2}) - \log \rho_{t_1}(X_{t_1})\,|\,X_0\right] = \mathbb{E}\left[\int_{t_1}^{t_2}\left(\frac{\partial g}{\partial t}(t, Y_t) + \frac{1}{2(1-t)}\text{tr}\left(\nabla^2 \log \rho_t(X_t)\right)\right)dt\,|\,X_0\right]. \tag{A.6}$$

We need the following lemmas, whose proof can be found at the end of this section.

***Claim*** 1. For any $0 < t < 1$ and any $y \in \mathbb{R}^d$, we have

$$\frac{\partial g}{\partial t}(t,y) = -\frac{d}{2t} + \frac{1}{2t^2}\int_{x_0}\rho_{X_0|X_t}\left(x_0\,|\,\sqrt{1-t}y\right)\|y - x_0\|_2^2 dx_0.$$

***Claim*** 2. For any $0 < t < 1$ and any $x \in \mathbb{R}^d$, we have

$$\text{tr}\left(\nabla^2 \log \rho_t(x)\right) = -\frac{d}{t} - \left\|\nabla \log \rho_t(x)\right\|_2^2 + \frac{1}{t^2}\int\left\|x - \sqrt{1-t}x_0\right\|_2^2 \rho_{X_0|X_t}\left(x_0\,|\,x\right)dx_0.$$

It also admits the lower bound

$$\text{tr}\left(\nabla^2 \log \rho_t(x)\right) \geq -\frac{d}{t}.$$

Therefore for any $x$ and $y = x/\sqrt{1-t}$, we know that

$$\frac{\partial g}{\partial t}(t, y) + \frac{1}{2(1-t)}\mathrm{tr}\left(\nabla^2 \log \rho_t(x)\right) \geq -\frac{d}{2t} - \frac{d}{2(1-t)t} \geq -\frac{d}{(1-t)t}. \tag{A.7}$$

Hence we have

$\mathbb{E}\left[\log \rho_{t_2}(X_{t_2}) - \log \rho_{t_1}(X_{t_1}) \mid X_0\right]$

$$\overset{\text{(i)}}{=} \mathbb{E}\left[\int_{t_1}^{t_2}\left(\frac{\partial g}{\partial t}(t, Y_t) + \frac{1}{2(1-t)}\mathrm{tr}\left(\nabla^2 \log \rho_t(X_t)\right) + \frac{d}{(1-t)t}\right)\mathrm{d}t \mid X_0\right] - \int_{t_1}^{t_2}\frac{d}{(1-t)t}\mathrm{d}t$$

$$\overset{\text{(ii)}}{=} \int_{t_1}^{t_2}\mathbb{E}\left[\left(\frac{\partial g}{\partial t}(t, Y_t) + \frac{1}{2(1-t)}\mathrm{tr}\left(\nabla^2 \log \rho_t(X_t)\right) + \frac{d}{(1-t)t}\right) \mid X_0\right]\mathrm{d}t - \int_{t_1}^{t_2}\frac{d}{(1-t)t}\mathrm{d}t$$

$$= \int_{t_1}^{t_2}\mathbb{E}\left[\left(\frac{\partial g}{\partial t}(t, Y_t) + \frac{1}{2(1-t)}\mathrm{tr}\left(\nabla^2 \log \rho_t(X_t)\right)\right) \mid X_0\right]\mathrm{d}t. \tag{A.8}$$

Here step (i) follows from (A.6), and its validity is guaranteed by

$$\int_{t_1}^{t_2}\frac{d}{t(1-t)}\mathrm{d}t = \log \frac{t_2(1-t_1)}{t_1(1-t_2)} < +\infty,$$

while step (ii) utilizes Tonelli's Theorem, and the nonnegativity of the integrand is ensured by (A.7). Taking Claims 1 and 2 collectively, we know that for any $x$ and $y = x/\sqrt{1-t}$,

$$\frac{\partial g}{\partial t}(t, y) - \frac{\mathrm{tr}\left(\nabla^2 \log \rho_t(x)\right)}{2(1-t)} = \frac{d + \left\|\nabla \log \rho_t(x)\right\|_2^2}{2(1-t)} + \frac{1}{2t^2}\int_{x_0}\rho_{X_0|X_t}\left(x_0 \mid \sqrt{1-t}y\right)\|y - x_0\|_2^2\mathrm{d}x_0$$

$$- \frac{1}{2(1-t)}\frac{1}{t^2}\int\left\|x - \sqrt{1-t}x_0\right\|_2^2\rho_{X_0|X_t}\left(x_0 \mid x\right)\mathrm{d}x_0$$

$$= \frac{d + \left\|\nabla \log \rho_t(x)\right\|_2^2}{2(1-t)}. \tag{A.9}$$

Putting (A.8) and (A.9) together, we arrive at

$$\mathbb{E}\left[\log \rho_{t_2}(X_{t_2}) - \log \rho_{t_1}(X_{t_1}) \mid X_0\right] = \int_{t_1}^{t_2}\mathbb{E}\left[\frac{d + \left\|\nabla \log \rho_t(X_t)\right\|_2^2}{2(1-t)} + \frac{1}{1-t}\mathrm{tr}\left(\nabla^2 \log \rho_t(X_t)\right) \mid X_0\right]\mathrm{d}t. \tag{A.10}$$

Notice that conditional on $X_0$, we have $X_t \sim \mathcal{N}(\sqrt{1-t}X_0, tI_d)$. Then we have

$\mathbb{E}\left[\log \rho_{t_2}(X_{t_2}) - \log \rho_{t_1}(X_{t_1}) \mid X_0\right]$

$$\overset{\text{(i)}}{=} \int_{t_1}^{t_2}\mathbb{E}\left[\frac{d + \left\|\nabla \log \rho_t(X_t)\right\|_2^2}{2(1-t)} + \frac{1}{1-t}\nabla \log \rho_t(X_t)^\top \frac{X_t - \sqrt{1-t}X_0}{t} \mid X_0\right]\mathrm{d}t$$

$$\overset{\text{(ii)}}{=} \int_{t_1}^{t_2}\left(\frac{1}{2(1-t)}\mathbb{E}\left[\left\|\frac{X_t - \sqrt{1-t}X_0}{t} + \nabla \log \rho_t(X_t)\right\|_2^2 \mid X_0\right] - \frac{d}{2t}\right)\mathrm{d}t$$

Here step (i) follows from (A.10) and an application of Stein's lemma

$$\mathbb{E}\left[\nabla \log \rho_t(X_t)^\top\left(X_t - \sqrt{1-t}X_0\right) \mid X_0\right] = t\mathbb{E}\left[\mathrm{tr}\left(\nabla^2 \log \rho_t(X_t)\right) \mid X_0\right],$$

while step (ii) holds since

$$\mathbb{E}\left[\left\|\frac{X_t - \sqrt{1-t}X_0}{t}\right\|_2^2\right] = \frac{d}{t}.$$

**Proof of Claim 1.** For any $t \in (0, 1)$, since $X_t = \sqrt{1-t}X_0 + \sqrt{t}Z$, we have

$$\rho_t(\sqrt{1-t}y) = \int_{x_0}(2\pi t)^{-d/2}\exp\left(-\frac{(1-t)\|y - x_0\|_2^2}{2t}\right)\rho_0(\mathrm{d}x_0). \tag{A.11}$$

Note that here $\rho_0(\cdot)$ stands for the law of $X_0$. Hence we have

$$
\begin{aligned}
\frac{\partial g}{\partial t}(t, y) &= \frac{\partial}{\partial t} \log \rho_t(\sqrt{1-t}y) = \frac{1}{\rho_t(\sqrt{1-t}y)} \frac{\partial}{\partial t} \rho_t(\sqrt{1-t}y) \\
&= \frac{1}{\rho_t(\sqrt{1-t}y)} \int_{x_0} (2\pi)^{-d/2} \Bigg[ -\frac{d}{2} t^{-d/2-1} \exp\Big( -\frac{(1-t)\|y-x_0\|_2^2}{2t} \Big) \\
&\qquad\qquad + t^{-d/2} \exp\Big( -\frac{(1-t)\|y-x_0\|_2^2}{2t} \Big) \frac{\|y-x_0\|_2^2}{2t^2} \Bigg] \rho_0(dx_0) \\
&= \frac{1}{\rho_t(\sqrt{1-t}y)} \int_{x_0} \rho_{X_t|X_0}\big(\sqrt{1-t}y \,|\, x_0\big) \Big[ -\frac{d}{2t} + \frac{\|y-x_0\|_2^2}{2t^2} \Big] \rho_0(dx_0) \\
&= \int_{x_0} \Big( -\frac{d}{2t} + \frac{\|y-x_0\|_2^2}{2t^2} \Big) \rho_{X_0|X_t}\big(dx_0 \,|\, \sqrt{1-t}y\big)
\end{aligned}
$$

as claimed.

**Proof of Claim 2.** Notice that we can express

$$
\nabla \log \rho_t(x) = -\frac{1}{t} \mathbb{E}\big[ X_t - \sqrt{1-t}X_0 \,|\, X_t = x \big] = -\frac{1}{t} \int_{x_0} \big(x - \sqrt{1-t}x_0\big) \rho_{X_0|X_t}(dx_0 \,|\, x);
$$

see Chen et al. (2022) for the proof of this relationship. Then we can compute

$$
\begin{aligned}
\nabla^2 \log \rho_t(x) &= -\frac{1}{t}\Big\{ I_d + \frac{1}{t}\mathbb{E}\big[ X_t - \sqrt{1-t}X_0 \,|\, X_t = x \big] \mathbb{E}\big[ X_t - \sqrt{1-t}X_0 \,|\, X_t = x \big]^\top \\
&\qquad\quad -\frac{1}{t}\mathbb{E}\Big[ \big(X_t - \sqrt{1-t}X_0\big)\big(X_t - \sqrt{1-t}X_0\big)^\top \,|\, X_t = x \Big] \Big\} \\
&= -\frac{1}{t}\Big\{ I_d + \frac{1}{t}\Big[ \int \big(x - \sqrt{1-t}x_0\big) \rho_{X_0|X_t}(dx_0\,|\,x) \Big]\Big[ \int \big(x - \sqrt{1-t}x_0\big) \rho_{X_0|X_t}(dx_0\,|\,x) \Big]^\top \\
&\qquad\quad -\frac{1}{t}\int \big(x - \sqrt{1-t}x_0\big)\big(x - \sqrt{1-t}x_0\big)^\top \rho_{X_0|X_t}(dx_0\,|\,x) \Big\}.
\end{aligned}
$$

Hence we have

$$
\begin{aligned}
\mathsf{tr}\big(\nabla^2 \log \rho_t(x)\big) &= -\frac{1}{t}\Big\{ d + \frac{1}{t}\Big\| \int \big(x - \sqrt{1-t}x_0\big) \rho_{X_0|X_t}(dx_0\,|\,x)\Big\|_2^2 - \frac{1}{t}\int \big\|x - \sqrt{1-t}x_0\big\|_2^2 \rho_{X_0|X_t}(dx_0\,|\,x) \Big\} \\
&= -\frac{d}{t} - \frac{1}{t^2}\big\|\nabla \log \rho_t(x)\big\|_2^2 + \frac{1}{t^2}\int \big\|x - \sqrt{1-t}x_0\big\|_2^2 \rho_{X_0|X_t}(x_0\,|\,x)\,dx_0.
\end{aligned}
$$

By Jensen's inequality, we know that

$$
\mathsf{tr}\big(\nabla^2 \log \rho_t(x)\big) \ge -\frac{d}{t}.
$$

# B  PROOF OF PROPOSITION 1

We establish the desired result by sandwiching $\mathbb{E}[\log \rho_t(X_t) \,|\, X_0 = x_0]$ and find its limit as $t \to 1$. We first record that the density of $X_t$ can be expressed as

$$
\rho_t(x) = \mathbb{E}_{X_0}\Big[ (2\pi t)^{-d/2} \exp\Big( -\frac{\|x - \sqrt{1-t}X_0\|_2^2}{2t} \Big) \Big], \tag{B.1}
$$

since $X_t \overset{\mathrm{d}}{=} \sqrt{1-t}X_0 + \sqrt{t}Z$ for an independent variable $Z \sim \mathcal{N}(0, I_d)$.

**Lower bounding $\mathbb{E}[\log \rho_t(X_t) \,|\, X_0 = x_0]$.** Starting from (B.1), for any $x \in \mathbb{R}^d$ and any $0 < t < 1$,

$$
\log \rho_t(x) = \log \mathbb{E}_{X_0}\Big[ (2\pi t)^{-d/2} \exp\Big( -\frac{\|x - \sqrt{1-t}X_0\|_2^2}{2t} \Big) \Big]
$$

$$\overset{(i)}{\geq} \log \left\{ (2\pi t)^{-d/2} \exp \left( - \mathbb{E}_{X_0} \left[ \frac{\|x - \sqrt{1-t}X_0\|_2^2}{2t} \right] \right) \right\}$$

$$= -\frac{d}{2}\log(2\pi t) - \mathbb{E}_{X_0}\left[ \frac{\|x - \sqrt{1-t}X_0\|_2^2}{2t} \right]$$

$$= -\frac{d}{2}\log(2\pi t) - \frac{\|x\|_2^2}{2t} - \frac{1-t}{2t}\mathbb{E}[\|X_0\|_2^2] + \frac{\sqrt{1-t}}{t}\mathbb{E}[x^\top X_0]$$

$$\overset{(ii)}{=} -\frac{d}{2}\log(2\pi t) - \left(1 + O(\sqrt{1-t})\right)\frac{\|x\|_2^2}{2t} + O(\sqrt{1-t})\mathbb{E}[\|X_0\|_2^2].$$

Here step (i) follows from Jensen's inequality and the fact that $e^{-x}$ is a convex function, while step (ii) follows from elementary inequalities

$$\left| \mathbb{E}[x^\top X_0] \right| \leq \mathbb{E}\left[ \|x\|\|X_0\|_2 \right] \leq \frac{1}{2}\mathbb{E}\left[ \|x\|_2^2 + \|X_0\|_2^2 \right].$$

This immediately gives, for any given $x_0 \in \mathbb{R}^d$ and any $0 < t < 1$,

$$\mathbb{E}[\log \rho_t(X_t) \,|\, X_0 = x_0] \geq \underbrace{-\frac{d}{2}\log(2\pi t) - \frac{1 + O(\sqrt{1-t})}{2t}\mathbb{E}\left[ \|X_t\|_2^2 \,|\, X_0 = x_0 \right] + O(\sqrt{1-t})\mathbb{E}[\|X_0\|_2^2]}_{=: f_{x_0}(t)}.$$
(B.2a)

Since $\mathbb{E}[\|X_0\|_2^2] < \infty$, it is straightforward to check that

$$\lim_{t \to 1-} f_{x_0}(t) = -\frac{d}{2}\log(2\pi) - \lim_{t \to 1-} \frac{1}{2}\mathbb{E}\left[ \|\sqrt{1-t}x_0 + \sqrt{t}Z\|_2^2 \right] \qquad \text{for } Z \sim \mathcal{N}(0, I_d)$$

$$= -\frac{d}{2}\log(2\pi) - \frac{d}{2}.$$
(B.2b)

**Upper bounding $\mathbb{E}[\log \rho_t(X_t) \,|\, X_0 = x_0]$.** Towards that, we need to obtain point-wise upper bound for $\log \rho_t(x)$. Since the desired result only depends on the limiting behavior when $t \to 1$, from now on we only consider $t > 0.9$, under which

$$(1-t)^{1/4} < \frac{1}{2}\sqrt{\log \frac{1}{1-t}}$$

holds. It would be helpful to develop the upper bound for the following two cases separately.

- For any $(1-t)^{1/4} < \|x\|_2 < 0.5\sqrt{\log 1/(1-t)}$, we have

$$\log \rho_t(x) \overset{(a)}{\leq} \log \mathbb{E}_{X_0}\left[ (2\pi t)^{-d/2} \exp\left( -\frac{(\|x\|_2 - (1-t)^{1/4})^2}{2t} \right) + \mathbb{1}\left( \|X_0\|_2 > (1-t)^{-1/4} \right) \right]$$

$$\overset{(b)}{\leq} -\frac{d}{2}\log(2\pi t) - \frac{(\|x\|_2 - (1-t)^{1/4})^2}{2t} + \exp\left( \frac{(\|x\|_2 - (1-t)^{1/4})^2}{2t} \right)\mathbb{P}\left( \|X_0\|_2 > (1-t)^{-1/4} \right)$$

$$\overset{(c)}{\leq} -\frac{d}{2}\log(2\pi t) - \frac{(\|x\|_2 - (1-t)^{1/4})^2}{2t} + \exp\left( \frac{\|x\|_2^2}{2t} \right)\mathbb{E}[\|X_0\|_2^2](1-t)^{1/2}$$

$$\overset{(d)}{\leq} -\frac{d}{2}\log(2\pi t) - \frac{(\|x\|_2 - (1-t)^{1/4})^2}{2t} + \mathbb{E}[\|X_0\|_2^2](1-t)^{1/4}.$$
(B.3)

Here step (a) follows from (B.1); step (b) holds since $\log(x + y) \leq \log x + y/x$ holds for any $x > 0$ and $y \geq 0$; step (c) follows from $\|x\|_2 > (1-t)^{1/4}$ and Chebyshev's inequality; while step (d) holds since $\|x\|_2 < 0.5\sqrt{\log 1/(1-t)}$.

- For $\|x\|_2 \geq 0.5\sqrt{\log 1/(1-t)}$ or $\|x\| \leq (1-t)^{1/4}$, we will use the naive upper bound

$$\log \rho_t(x) \leq -\frac{d}{2}\log(2\pi t) < 0,$$
(B.4)

where the first relation simply follows from (B.1) and the second relation holds when $t > 0.9$.

Then we have

$$\mathbb{E}[\log \rho_t(X_t) \mid X_0 = x_0] \overset{\text{(i)}}{\leq} \mathbb{E}[\log \rho_t(X_t) \, \mathbb{1}\left\{(1-t)^{1/4} < \|X_t\|_2 < 0.5\sqrt{\log 1/(1-t)}\right\} \mid X_0 = x_0]$$

$$\overset{\text{(ii)}}{\leq} \mathbb{E}\left[\left(-\frac{d}{2}\log(2\pi t) - \frac{(\|x\|_2 - (1-t)^{1/4})^2}{2t} + \mathbb{E}[\|X_0\|_2^2](1-t)^{1/4}\right)\right.$$

$$\left. \cdot \mathbb{1}\left\{(1-t)^{1/4} < \|X_t\|_2 < 0.5\sqrt{\log 1/(1-t)}\right\} \mid X_0 = x_0\right]$$

$$= \underbrace{\left(-\frac{d}{2}\log(2\pi t) + \mathbb{E}[\|X_0\|_2^2](1-t)^{1/4}\right)\mathbb{P}\left((1-t)^{1/4} < \|X_t\|_2 < 0.5\sqrt{\log 1/(1-t)}\right)}_{=:\overline{g}_{x_0}(t)}$$

$$- \underbrace{\mathbb{E}\left[\frac{(\|X_t\|_2 - (1-t)^{1/4})^2}{2t}\mathbb{1}\left\{(1-t)^{1/4} < \|X_t\|_2 < 0.5\sqrt{\log 1/(1-t)}\right\} \mid X_0 = x_0\right]}_{=:\widetilde{g}_{x_0}(t)}.$$

Here step (i) follows from (B.4), while step (ii) utilizes (B.3). Since $X_t$ is a continuous random variable for any $t \in (0,1)$, we have

$$\lim_{t \to 1-} \mathbb{P}\left((1-t)^{1/4} < \|X_t\|_2 < 0.5\sqrt{\log 1/(1-t)}\right) = 1.$$

Therefore we know that

$$\lim_{t \to 1-} \overline{g}_{x_0}(t) = -\frac{d}{2}\log(2\pi).$$

Recall that $X_t \overset{\text{d}}{=} \sqrt{1-t}X_0 + \sqrt{t}Z$ for a Gaussian variable $Z \sim \mathcal{N}(0, I_d)$ independent of $X_0$, we can express

$$\widetilde{g}_{x_0}(t) = \mathbb{E}\left[\frac{(\|\sqrt{t}Z + \sqrt{1-t}x_0\|_2 - (1-t)^{1/4})^2}{2t}\mathbb{1}\left\{(1-t)^{1/4} < \|\sqrt{t}Z + \sqrt{1-t}x_0\|_2 < \frac{1}{2}\sqrt{\log 1/(1-t)}\right\}\right]$$

$$= \int \underbrace{\frac{(\|\sqrt{t}z + \sqrt{1-t}x_0\|_2 - (1-t)^{1/4})^2}{2t}\mathbb{1}\left\{(1-t)^{1/4} < \|\sqrt{t}z + \sqrt{1-t}x_0\|_2 < \frac{1}{2}\sqrt{\log \frac{1}{1-t}}\right\}\phi(z)}_{=:h_t(z)}\,\mathrm{d}z,$$

where $\phi(z) = (2\pi)^{-d/2}\exp(-\|z\|_2^2/2)$ is the density function of $\mathcal{N}(0, I_d)$. For any $t \in (0.9, 1)$, we have

$$h_t(z) \leq \|\sqrt{t}z + \sqrt{1-t}x_0\|_2^2\phi(z) \leq 2(\|z\|_2^2 + \|x_0\|_2^2)\phi(z) =: h(z),$$

and it is straightforward to check that

$$\int h(z)\mathrm{d}z = 2d + 2\|x_0\|_2^2 < \infty.$$

By dominated convergence theorem, we know that

$$\lim_{t \to 1-} \widetilde{g}_{x_0}(t) = \lim_{t \to 1-} \int h_t(z)\mathrm{d}z = \int \lim_{t \to 1-} h_t(z)\mathrm{d}z = \int \frac{\|z\|_2^2}{2}\phi(z)\mathrm{d}z = \frac{d}{2}.$$

Therefore we have

$$\mathbb{E}[\log \rho_t(X_t) \mid X_0 = x_0] \leq g_{x_0}(t) \qquad \text{where} \qquad g_{x_0}(t) := \overline{g}_{x_0}(t) - \widetilde{g}_{x_0}(t), \qquad (\text{B.5a})$$

such that

$$\lim_{t \to 1-} g_{x_0}(t) = \lim_{t \to 1-} \overline{g}_{x_0}(t) - \lim_{t \to 1-} \widetilde{g}_{x_0}(t) = -\frac{d}{2}\log(2\pi) - \frac{d}{2}. \qquad (\text{B.5b})$$

**Conclusion.** By putting together (B.2) and (B.5), we know that for any $t \in (0.9, 1)$

$$f_{x_0}(t) \leq \mathbb{E}[\log \rho_t(X_t) \mid X_0 = x_0] \leq g_{x_0}(t) \qquad \text{and} \qquad \lim_{t \to 1-} f_{x_0}(t) = \lim_{t \to 1-} g_{x_0}(t) = -\frac{d}{2}\log(2\pi) - \frac{d}{2}.$$

By the sandwich theorem, we arrive at the desired result

$$\lim_{t \to 1-} \mathbb{E}[\log \rho_t(X_t) \mid X_0 = x_0] = -\frac{d}{2}\log(2\pi) - \frac{d}{2}.$$

## C  PROOF OF PROPOSITION 2

Suppose that $L := \sup_x \|\nabla^2 \log \rho_0(x)\|$. The following claim will be useful in establishing the proposition, whose proof is deferred to the end of this section.

***Claim* 3.** There exists some $t_0 > 0$ such that

$$\sup_x \|\nabla^2 \log \rho_t(x)\| \leq 4L. \tag{C.1}$$

holds for any $0 \leq t \leq t_0$.

Equipped with Claim 3, we know that for any $t \leq t_0$,

$$
\begin{aligned}
\mathbb{E}\big[\log \rho_t(X_t) \,|\, X_0 = x_0\big] &= \mathbb{E}\big[\log \rho_t(\sqrt{1-t}x_0 + \sqrt{t}Z)\big] \\
&\overset{\text{(i)}}{=} \mathbb{E}\big[\log \rho_t(\sqrt{1-t}x_0) + \sqrt{t}Z^\top \nabla \log \rho_t(\sqrt{1-t}x_0) + O(Lt)\|Z\|_2^2\big] \\
&\overset{\text{(ii)}}{=} \log \rho_t(\sqrt{1-t}x_0) + O(Ldt) \\
&\overset{\text{(iii)}}{=} \log \int_x \rho_0(x)(2\pi t)^{-d/2} \exp\Big(-\frac{(1-t)\|x-x_0\|_2^2}{2t}\Big)\mathrm{d}x + O(L\sqrt{dt}) \\
&= (1-t)^{-d/2} \log \int_x \rho_0(x)\Big(\frac{2\pi t}{1-t}\Big)^{-d/2} \exp\Big(-\frac{(1-t)\|x-x_0\|_2^2}{2t}\Big)\mathrm{d}x + O(L\sqrt{dt}),
\end{aligned}
\tag{C.2}
$$

where $Z \sim \mathcal{N}(0, I_d)$. Here step (i) follows from (C.1) in Claim 3; step (ii) holds since $\mathbb{E}[Z] = 0$ and $\mathbb{E}[\|Z\|_2^2] = d$; while step (iii) follows from (A.11). It is straightforward to check that

$$\int_x \rho_0(x)\Big(\frac{2\pi t}{1-t}\Big)^{-d/2} \exp\Big(-\frac{(1-t)\|x-x_0\|_2^2}{2t}\Big)\mathrm{d}x$$

is the density of $\rho_0 * \mathcal{N}(0, t/(1-t))$ evaluated at $x_0$, which taken collectively with the assumption that $\rho_0(\cdot)$ is continuous yields

$$\lim_{t\to 0+} \int_x \rho_0(x)\Big(\frac{2\pi t}{1-t}\Big)^{-d/2} \exp\Big(-\frac{(1-t)\|x-x_0\|_2^2}{2t}\Big)\mathrm{d}x = \rho_0(x_0).$$

Therefore we can take $t \to 0+$ in (C.2) to achieve

$$\lim_{t\to 0+} \mathbb{E}\big[\log \rho_t(X_t) \,|\, X_0 = x_0\big] = \log \rho_0(x_0)$$

as claimed.

**Proof of Claim 3.**  The conditional density of $X_0$ given $X_t = x$ is

$$p_{X_0|X_t}(x_0\,|\,x) = \frac{p_{X_0}(x_0)p_{X_t|X_0}(x\,|\,x_0)}{p_{X_t}(x)} = \frac{\rho_0(x_0)}{\rho_t(x)}(2\pi t)^{-d/2}\exp\Big(-\frac{\|x-\sqrt{1-t}x_0\|_2^2}{2t}\Big), \tag{C.3}$$

which leads to

$$
\begin{aligned}
-\nabla_{x_0}^2 \log p_{X_0|X_t}(x_0\,|\,x) &= -\nabla_{x_0}^2 \log \rho_0(x_0) + \frac{1}{2t}\nabla_{x_0}^2 \|x-\sqrt{1-t}x_0\|_2^2 \\
&= -\nabla_{x_0}^2 \log \rho_0(x_0) + \frac{1-t}{t}I_d \succeq \Big(\frac{1-t}{t} - L\Big)I_d.
\end{aligned}
$$

Therefore we know that

$$-\nabla_{x_0}^2 \log p_{X_0|X_t}(x_0\,|\,x) \succeq \frac{1}{2t}I_d \qquad \text{for} \qquad t \leq \frac{1}{2(L+1)}, \tag{C.4}$$

namely the conditional distribution of $X_0$ given $X_t = x$ is $1/(2t)$-strongly log-concave for any $x$, when $t \leq 1/2(L+1)$. By writting

$$\rho_t(x) = p_{X_t}(x) = \int \phi(z)p_{\sqrt{1-t}X_0}\Big(x - \sqrt{t}z\Big)\mathrm{d}z = (1-t)^{-d/2}\int \phi(z)\rho_0\Big(\frac{x-\sqrt{t}z}{\sqrt{1-t}}\Big)\mathrm{d}z, \tag{C.5}$$

we can express the score function of $\rho_t$ as

$$
\nabla \log \rho_t(x) = \frac{\nabla \rho_t(x)}{\rho_t(x)} = (1-t)^{-\frac{d+1}{2}} \frac{1}{\rho_t(x)} \int \phi(z) \nabla \rho_0 \left( \frac{x - \sqrt{t}z}{\sqrt{1-t}} \right) \mathrm{d}z
$$

$$
= (1-t)^{-\frac{d+1}{2}} \frac{1}{\rho_t(x)} \int \phi(z) \rho_0 \left( \frac{x - \sqrt{t}z}{\sqrt{1-t}} \right) \nabla \log \rho_0 \left( \frac{x - \sqrt{t}z}{\sqrt{1-t}} \right) \mathrm{d}z \tag{C.6}
$$

$$
\overset{\text{(i)}}{=} (1-t)^{-\frac{d+1}{2}} \left( \frac{1-t}{t} \right)^{d/2} \frac{1}{\rho_t(x)} \int \phi \left( \frac{x - \sqrt{1-t}x_0}{\sqrt{t}} \right) \rho_0(x_0) \nabla \log \rho_0(x_0) \mathrm{d}x_0
$$

$$
\overset{\text{(ii)}}{=} \frac{1}{\sqrt{1-t}} \int p_{X_0 | X_t}(x_0 \mid x) \nabla \log \rho_0(x_0) \mathrm{d}x_0 = \frac{1}{\sqrt{1-t}} \mathbb{E} \left[ \nabla \log \rho_0(X_0) \mid X_t = x \right]. \tag{C.7}
$$

Here step (i) uses the change of variable $x_0 = (x - \sqrt{t}z)/\sqrt{1-t}$, while step (ii) follows from (C.3). Starting from (C.6), we take the derivative to achieve

$$
\nabla^2 \log \rho_t(x) = \underbrace{(1-t)^{-\frac{d}{2}+1} \frac{1}{\rho_t(x)} \int \phi(z) \rho_0 \left( \frac{x - \sqrt{t}z}{\sqrt{1-t}} \right) \nabla \log \rho_0 \left( \frac{x - \sqrt{t}z}{\sqrt{1-t}} \right) \left[ \nabla \log \rho_0 \left( \frac{x - \sqrt{t}z}{\sqrt{1-t}} \right) \right]^\top \mathrm{d}z}_{=:H_1(x)}
$$

$$
+ \underbrace{(1-t)^{-\frac{d}{2}+1} \frac{1}{\rho_t(x)} \int \phi(z) \rho_0 \left( \frac{x - \sqrt{t}z}{\sqrt{1-t}} \right) \nabla^2 \log \rho_0 \left( \frac{x - \sqrt{t}z}{\sqrt{1-t}} \right) \mathrm{d}z}_{=:H_2(x)}
$$

$$
- \underbrace{(1-t)^{-\frac{d+1}{2}} \frac{1}{\rho_t^2(x)} \int \phi(z) \rho_0 \left( \frac{x - \sqrt{t}z}{\sqrt{1-t}} \right) \nabla \log \rho_0 \left( \frac{x - \sqrt{t}z}{\sqrt{1-t}} \right) \mathrm{d}z \left[ \nabla \rho_t(x) \right]^\top}_{=:H_3(x)}. \tag{C.8}
$$

Then we investigate $H_1(x)$, $H_2(x)$ and $H_3(x)$ respectively. Regarding $H_1(x)$, we have

$$
H_1(x) \overset{\text{(a1)}}{=} (1-t)^{-\frac{d}{2}+1} \left( \frac{1-t}{t} \right)^{d/2} \frac{1}{\rho_t(x)} \int \phi \left( \frac{x - \sqrt{1-t}x_0}{\sqrt{t}} \right) \rho_0(x_0) \nabla \log \rho_0(x_0) \left[ \nabla \log \rho_0(x_0) \right]^\top \mathrm{d}z
$$

$$
\overset{\text{(b1)}}{=} \frac{1}{1-t} \int p_{X_0 | X_t}(x_0 \mid x) \nabla \log \rho_0(x_0) \left[ \nabla \log \rho_0(x_0) \right]^\top \mathrm{d}x_0
$$

$$
= \frac{1}{1-t} \mathbb{E} \left[ \nabla \log \rho_0(X_0) \left[ \nabla \log \rho_0(X_0) \right]^\top \mid X_t = x \right]; \tag{C.9a}
$$

for $H_2(x)$, we have

$$
H_2(x) \overset{\text{(a2)}}{=} (1-t)^{-\frac{d}{2}+1} \left( \frac{1-t}{t} \right)^{d/2} \frac{1}{\rho_t(x)} \int \phi \left( \frac{x - \sqrt{1-t}x_0}{\sqrt{t}} \right) \rho_0(x_0) \nabla^2 \log \rho_0 \left( \frac{x - \sqrt{t}z}{\sqrt{1-t}} \right) \mathrm{d}x_0
$$

$$
\overset{\text{(b2)}}{=} \frac{1}{1-t} \int p_{X_0 | X_t}(x_0 \mid x) \nabla^2 \log \rho_0(x_0) \mathrm{d}x_0 = \frac{1}{1-t} \mathbb{E} \left[ \nabla^2 \log \rho_0(X_0) \mid X_t = x \right]; \tag{C.9b}
$$

for the final term $H_3(x)$, we have

$$
H_3(x) \overset{\text{(c)}}{=} -(1-t)^{-\frac{d+1}{2}} \frac{1}{\rho_t(x)} \left[ \int \phi(z) \rho_0 \left( \frac{x - \sqrt{t}z}{\sqrt{1-t}} \right) \nabla \log \rho_0 \left( \frac{x - \sqrt{t}z}{\sqrt{1-t}} \right) \mathrm{d}z \right] \left[ \nabla \log \rho_t(x) \right]^\top
$$

$$
\overset{\text{(a3)}}{=} -(1-t)^{-\frac{d+1}{2}} \left( \frac{1-t}{t} \right)^{d/2} \frac{1}{\rho_t(x)} \left[ \int \phi \left( \frac{x - \sqrt{1-t}x_0}{\sqrt{t}} \right) \rho_0(x_0) \nabla \log \rho_0(x_0) \mathrm{d}x_0 \right] \left[ \nabla \log \rho_t(x) \right]^\top
$$

$$
\overset{\text{(b3)}}{=} -\frac{1}{\sqrt{1-t}} \int p_{X_0 | X_t}(x_0 \mid x) \nabla \log \rho_0(x_0) \mathrm{d}x_0 \left[ \nabla \log \rho_t(x) \right]^\top
$$

$$
\overset{\text{(d)}}{=} -\frac{1}{1-t} \mathbb{E} \left[ \nabla \log \rho_0(X_0) \mid X_t = x \right] \mathbb{E} \left[ \nabla \log \rho_0(X_0) \mid X_t = x \right]^\top. \tag{C.9c}
$$

Here steps (a1), (a2) and (a3) follow from the change of variable $x_0 = (x - \sqrt{t}z)/\sqrt{1-t}$; steps (b1), (b2) and (b3) utilize (C.3); step (c) follows from $\nabla \log \rho_t(x) = \nabla \rho_t(x)/\rho_t(x)$; while step (d) follows from (C.7). Substituting (C.9) back into (C.8), we have

$$\nabla^2 \log \rho_t(x) = \frac{1}{1-t} \mathbb{E}\left[\nabla^2 \log \rho_0(X_0) \mid X_t = x\right] + \frac{1}{1-t} \mathsf{cov}\left(\nabla \log \rho_0(X_0) \mid X_t = x\right). \quad \text{(C.10)}$$

Notice that for any $t \leq 1/2(L+1)$, we have

$$\|\mathsf{cov}\left(\nabla \log \rho_0(X_0) \mid X_t = x\right)\| = \sup_{u \in \mathbb{S}^{d-1}} \mathbb{E}\left[\left[u^\top \left(\nabla \log \rho_0(X_0) - \mathbb{E}\left[\nabla \log \rho_0(X_0) \mid X_t = x\right]\right)\right]^2 \mid X_t = x\right]$$

$$\overset{\text{(i)}}{\leq} \sup_{u \in \mathbb{S}^{d-1}} \mathbb{E}\left[\left[u^\top \left(\nabla \log \rho_0(X_0) - \nabla \log \rho_0\left(\mathbb{E}\left[X_0 \mid X_t = x\right]\right)\right)\right]^2 \mid X_t = x\right]$$

$$\leq \mathbb{E}\left[\|\nabla \log \rho_0(X_0) - \nabla \log \rho_0\left(\mathbb{E}\left[X_0 \mid X_t = x\right]\right)\|_2^2 \mid X_t = x\right]$$

$$\overset{\text{(ii)}}{\leq} \mathbb{E}\left[\|X_0 - \mathbb{E}\left[X_0 \mid X_t = x\right]\|_2^2 \mid X_t = x\right]$$

$$\overset{\text{(iii)}}{\leq} 2tL^2 d, \quad \text{(C.11)}$$

Here step (i) holds since for any random variable $X$, $\mathbb{E}[(X-c)^2]$ is minimized at $c = \mathbb{E}[X]$; step (ii) holds since the score function $\nabla \log \rho_0(\cdot)$ is $L$-Lipschitz; step (iii) follows from the Poincaré inequality for log-concave distribution, and the fact that the conditional distribution of $X_0$ given $X_t = x$ is $1/2t$-strongly log-concave (cf. (C.4)). We conclude that

$$\|\nabla^2 \log \rho_t(x)\| \overset{\text{(a)}}{\leq} \frac{1}{1-t}L + \frac{2tL^2 d}{1-t} \overset{\text{(b)}}{\leq} 4L.$$

Here step (a) follows from (C.10), (C.11), and the assumption that $\sup_x \|\nabla^2 \log \rho_t(x)\| \leq L$, while step (b) holds provided that $t \leq \min\{1/2, 1/(2Ld)\}$.

# D   MORE DISCUSSIONS ON THE DENSITY FORMULAS

Although the density formulas (3.1a) have been rigorously established, it is helpful to inspect the limiting behavior of the integrand $D(t, x_0)$ at the boundary to understand why the integral converges. Throughout the discussion, we let $\varepsilon \sim \mathcal{N}(0, I_d)$.

- As $t \to 0$, we can compute

$$D(t, x_0) \asymp \frac{\mathbb{E}\left[\|\varepsilon + \sqrt{t}\nabla \log \rho_t(\sqrt{1-t}x_0 + \sqrt{t}\varepsilon)\|_2^2\right] - d}{t}$$

$$\overset{\text{(i)}}{\asymp} \mathbb{E}\left[\|\nabla \log \rho_t(\sqrt{1-t}x_0 + \sqrt{t}\varepsilon)\|_2^2\right] + \frac{1}{\sqrt{t}}\mathbb{E}\left[\varepsilon^\top \nabla \log \rho_t(\sqrt{1-t}x_0 + \sqrt{t}\varepsilon)\right]$$

$$\overset{\text{(ii)}}{\asymp} \mathbb{E}\left[\|\nabla \log \rho_t(\sqrt{1-t}x_0 + \sqrt{t}\varepsilon)\|_2^2\right] + \mathbb{E}\left[\mathsf{tr}\left(\nabla^2 \log \rho_t(\sqrt{1-t}x_0 + \sqrt{t}\varepsilon)\right)\right].$$

  Here step (i) holds since $\mathbb{E}[\|\varepsilon\|_2^2] = d$, while step (ii) follows from Stein's lemma. Therefore, when the score functions are reasonably smooth as $t \to 0$, one may expect that the integrand $D(t, x_0)$ is of constant order, allowing the integral to converge at $t = 0$.

- As $t \to 1$, we can compute

$$D(t, x_0) = \frac{1}{2(1-t)t}\mathbb{E}\left[\|\varepsilon + \sqrt{t}\nabla \log \rho_t(\sqrt{1-t}x_0 + \sqrt{t}\varepsilon)\|_2^2\right] - \frac{d}{2t}$$

$$\asymp \frac{1}{2(1-t)}\mathbb{E}\left[\|\varepsilon + \sqrt{t}\nabla \log \rho_t(\sqrt{1-t}x_0 + \sqrt{t}\varepsilon)\|_2^2\right] - \frac{d}{2}.$$

  Since $\rho_t$ converges to $\phi$ as $t \to 1$ and $\nabla \log \phi(x) = -x$, we have

$$\lim_{t \to 1} \varepsilon + \sqrt{t}\nabla \log \rho_t(\sqrt{1-t}x_0 + \sqrt{t}\varepsilon) = 0.$$

  Hence one may expect that $\mathbb{E}\left[\|\varepsilon + \sqrt{t}\nabla \log \rho_t(\sqrt{1-t}x_0 + \sqrt{t}\varepsilon)\|_2^2\right]$ converges to zero quickly, allowing the integral to converge at $t = 1$.

# E   TECHNICAL DETAILS IN SECTION 4

## E.1   TECHNICAL DETAILS IN SECTION 4.1

**Computing $L_{t-1}(x_0)$.**   Conditional on $X_t = x_t$ and $X_0 = x_0$, we have

$$X_{t-1} \mid X_t = x_t, X_0 = x_0 \sim \mathcal{N}\left( \frac{\sqrt{\overline{\alpha}_{t-1}}\beta_t}{1-\overline{\alpha}_t}x_0 + \frac{\sqrt{\alpha_t}(1-\overline{\alpha}_{t-1})}{1-\overline{\alpha}_t}x_t, \frac{1-\overline{\alpha}_{t-1}}{1-\overline{\alpha}_t}\beta_t I_d \right),$$

and conditional on $Y_t = x_t$, we have

$$Y_{t-1} \mid Y_t = x_t \sim \mathcal{N}\left( \frac{x_t + \eta_t s_t(x_t)}{\sqrt{\alpha_t}}, \frac{\sigma_t^2}{\alpha_t} \right).$$

Recall that the KL divergence between two $d$-dimensional Gaussian $\mathcal{N}(\mu_1, \Sigma_1)$ and $\mathcal{N}(\mu_2, \Sigma_2)$ admits the following closed-form expression:

$$\mathsf{KL}\left( \mathcal{N}(\mu_1, \Sigma_1) \, \| \, \mathcal{N}(\mu_2, \Sigma_2) \right) = \frac{1}{2}\left[ \mathsf{tr}\left( \Sigma_2^{-1}\Sigma_1 \right) + (\mu_2 - \mu_1)^\top \Sigma_2^{-1}(\mu_2 - \mu_1) - d + \log \det \Sigma_2 - \log \det \Sigma_1 \right].$$

Then we can check that for $2 \le t \le T$,

$$\mathsf{KL}\left( p_{X_{t-1} \mid X_t, X_0}(\cdot \mid x_t, x_0) \, \| \, p_{Y_{t-1} \mid Y_t}(\cdot \mid x_t) \right) = \frac{\alpha_t}{2\sigma_t^2}\left\| \frac{\sqrt{\overline{\alpha}_{t-1}}\beta_t}{1-\overline{\alpha}_t}x_0 + \frac{\alpha_t - 1}{\sqrt{\alpha_t}(1-\overline{\alpha}_t)}x_t - \frac{\eta_t s_t(x_t)}{\sqrt{\alpha_t}} \right\|_2^2,$$

where we use the coefficient design (4.3). This immediately gives

$$
\begin{aligned}
L_{t-1}(x_0) &= \frac{\alpha_t}{2\sigma_t^2}\mathbb{E}_{x_t \sim p_{X_t \mid X_0}(\cdot \mid x_0)}\left[ \left\| \frac{\sqrt{\overline{\alpha}_{t-1}}\beta_t}{1-\overline{\alpha}_t}x_0 + \frac{\alpha_t - 1}{\sqrt{\alpha_t}(1-\overline{\alpha}_t)}x_t - \frac{\eta_t s_t(x_t)}{\sqrt{\alpha_t}} \right\|_2^2 \right] \\
&\stackrel{\text{(i)}}{=} \frac{\alpha_t}{2\sigma_t^2}\mathbb{E}_{\varepsilon \sim \mathcal{N}(0,I_d)}\left[ \left\| \frac{\alpha_t - 1}{\sqrt{\alpha_t}(1-\overline{\alpha}_t)}\varepsilon - \frac{1-\alpha_t}{\sqrt{\alpha_t}}s_t(\sqrt{\overline{\alpha}_t}x_0 + \sqrt{1-\overline{\alpha}_t}\varepsilon) \right\|_2^2 \right] \\
&\stackrel{\text{(ii)}}{=} \frac{1-\alpha_t}{2(\alpha_t - \overline{\alpha}_t)}\mathbb{E}_{\varepsilon \sim \mathcal{N}(0,I_d)}\left[ \left\| \varepsilon - \varepsilon_t(\sqrt{\overline{\alpha}_t}x_0 + \sqrt{1-\overline{\alpha}_t}\varepsilon) \right\|_2^2 \right].
\end{aligned}
$$

Here in step (i), we utilize the coefficient design (4.3) and replace $x_t$ with $\sqrt{\overline{\alpha}_t}x_0 + \sqrt{1-\overline{\alpha}_t}\varepsilon$, which has the same distribution; while in step (ii), we replace the score function $s_t(\cdot)$ with the epsilon predictor $\varepsilon_t(\cdot) := -\sqrt{1-\overline{\alpha}_t}s_t(\cdot)$. Comparing the coefficients in $L_{t-1}^\star$ and $L_{t-1}$, we decompose

$$\left| \frac{1-\alpha_{t+1}}{2(1-\overline{\alpha}_t)} - \frac{1-\alpha_t}{2(\alpha_t - \overline{\alpha}_t)} \right| \le \underbrace{\left| \frac{1-\alpha_{t+1}}{2(1-\overline{\alpha}_t)} - \frac{1-\alpha_{t+1}}{2(\alpha_t - \overline{\alpha}_t)} \right|}_{=:\gamma_1} + \underbrace{\left| \frac{1-\alpha_{t+1}}{2(\alpha_t - \overline{\alpha}_t)} - \frac{1-\alpha_t}{2(\alpha_t - \overline{\alpha}_t)} \right|}_{=:\gamma_2}.$$

Consider the learning rate schedule in Li et al. (2023b); Li & Yan (2024):

$$\beta_1 = \frac{1}{T^{c_0}}, \qquad \beta_{t+1} = \frac{c_1 \log T}{T} \min\left\{ \beta_1\left( 1 + \frac{c_1 \log T}{T} \right)^t, 1 \right\} \quad (t = 1, \ldots, T-1) \quad \text{(E.1)}$$

for sufficiently large constants $c_0, c_1 > 0$. Then using the properties in e.g., Li & Yan (2024, Lemma 8), we can check that

$$\gamma_1 = \left| \frac{(1-\alpha_{t+1})(\alpha_t - 1)}{2(1-\overline{\alpha}_t)(\alpha_t - \overline{\alpha}_t)} \right| \le \frac{8c_1 \log T}{T}\left| \frac{1-\alpha_{t+1}}{2(1-\overline{\alpha}_t)} \right|,$$

and

$$\gamma_2 = \left| \frac{\alpha_t - \alpha_{t+1}}{2(\alpha_t - \overline{\alpha}_t)} \right| = \left| \frac{\beta_t - \beta_{t+1}}{2(\alpha_t - \overline{\alpha}_t)} \right| \le \left| 1 - \frac{\beta_t}{\beta_{t+1}} \right| \left| 1 + \frac{1-\alpha_t}{\alpha_t - \overline{\alpha}_t} \right| \left| \frac{1-\alpha_{t+1}}{2(1-\overline{\alpha}_t)} \right| \le \frac{8c_1 \log T}{T}\left| \frac{1-\alpha_{t+1}}{2(1-\overline{\alpha}_t)} \right|.$$

Hence the coefficients in $L_{t-1}^\star$ and $L_{t-1}$ are identical up to higher-order error:

$$\left| \frac{1-\alpha_{t+1}}{2(1-\overline{\alpha}_t)} - \frac{1-\alpha_t}{2(\alpha_t - \overline{\alpha}_t)} \right| \le \frac{16c_1 \log T}{T}\left| \frac{1-\alpha_{t+1}}{2(1-\overline{\alpha}_t)} \right|.$$

**Computing $L_0(x_0)$.** By taking $\eta_1 = \sigma_1^2 = 1 - \alpha_1$ (notice that (4.3) does not cover the case $t = 1$), we have

$$p_{Y_0|Y_1}(x_0 \,|\, x_1) = \left(\frac{2\pi\sigma_1^2}{\alpha_1}\right)^{-d/2} \exp\left(-\frac{\alpha_1}{2\sigma_1^2}\left\|x_0 - \frac{x_1 - \eta_1 s_1(x_1)}{\sqrt{\alpha_1}}\right\|_2^2\right)$$

$$= \left(\frac{2\pi\beta_1}{\alpha_1}\right)^{-d/2} \exp\left(-\frac{\alpha_1}{2\beta_1}\left\|x_0 - \frac{x_1 - \beta_1 s_1(x_1)}{\sqrt{\alpha_1}}\right\|_2^2\right),$$

and therefore

$$C_0(x_0) = \mathbb{E}_{x_1 \sim p_{X_1|X_0}(\cdot\,|\,x_0)} \left[-\frac{d}{2}\log\frac{2\pi\beta_1}{\alpha_1} - \frac{\alpha_1}{2\beta_1}\left\|x_0 - \frac{x_1 + \beta_1 s_1(x_1)}{\sqrt{\alpha_1}}\right\|_2^2\right]$$

$$\overset{\text{(i)}}{=} -\frac{d}{2}\log\frac{2\pi\beta_1}{\alpha_1} - \frac{1}{2}\mathbb{E}_{\varepsilon\sim\mathcal{N}(0,I_d)}\left[\|\varepsilon + \sqrt{\beta_1}s_1(\sqrt{1-\beta_1}x_0 + \sqrt{\beta_1}\varepsilon)\|_2^2\right]$$

$$\overset{\text{(ii)}}{=} -\frac{1+\log(2\pi\beta_1)}{2}d + \frac{d}{2}\log(1-\beta_1) - \frac{1}{2}\beta_1\mathbb{E}_{\varepsilon\sim\mathcal{N}(0,I_d)}\left[\|s_1(\sqrt{1-\beta_1}x_0 + \sqrt{\beta_1}\varepsilon)\|_2^2\right]$$

$$- \sqrt{\beta_1}\mathbb{E}_{\varepsilon\sim\mathcal{N}(0,I_d)}\left[\varepsilon^\top s_1(\sqrt{1-\beta_1}x_0 + \sqrt{\beta_1}\varepsilon)\right]. \tag{E.2}$$

Here in step (i), we replace $x_1$ with $\sqrt{1-\beta_1}x_0 + \sqrt{\beta_1}\varepsilon$, which has the same distribution; step (ii) uses the fact that $\mathbb{E}[\|\varepsilon\|_2^2] = d$ for $\varepsilon \sim \mathcal{N}(0, I_d)$. Using similar analysis as in Proposition 2, we can show that $\sup_x \|\nabla^2 \log q_1(x)\| \leq O(L)$ when $\beta_1$ is sufficiently small, as long as $\sup_x \|\nabla^2 \log q_0(x)\| \leq L$. Hence we have

$$\mathbb{E}_{\varepsilon\sim\mathcal{N}(0,I_d)}\left[\|s_1(\sqrt{1-\beta_1}x_0 + \sqrt{\beta_1}\varepsilon)\|_2^2\right] \leq \mathbb{E}_{\varepsilon\sim\mathcal{N}(0,I_d)}\left[\left(\|s_1(x_0)\|_2 + O(L)\|x_0 - \sqrt{1-\beta_1}x_0 - \sqrt{\beta_1}\varepsilon\|_2\right)^2\right]$$

$$\leq 2\|s_1(x_0)\|_2^2 + O(L^2)\mathbb{E}_{\varepsilon\sim\mathcal{N}(0,I_d)}\left[\|x_0 - \sqrt{1-\beta_1}x_0 - \sqrt{\beta_1}\varepsilon\|_2^2\right]$$

$$= 2\|s_1(x_0)\|_2^2 + O(L^2\beta_1). \tag{E.3}$$

By Stein's lemma, we can show that

$$\mathbb{E}_{\varepsilon\sim\mathcal{N}(0,I_d)}\left[\varepsilon^\top s_1(\sqrt{1-\beta_1}x_0 + \sqrt{\beta_1}\varepsilon)\right] = \sqrt{\beta_1}\mathbb{E}\left[\mathsf{tr}\left(\nabla^2 \log q_1(\sqrt{1-\beta_1}x_0 + \sqrt{\beta_1}\varepsilon)\right)\right]$$

$$\leq O(\sqrt{\beta_1}Ld). \tag{E.4}$$

Substituting the bounds (E.3) and (E.4) back into (E.2), we have

$$C_0(x_0) = -\frac{1+\log(2\pi\beta_1)}{2}d + O(\beta_1)$$

as claimed.

**Negligibility of $L_T(x)$.** Since

$$Y_T \sim \mathcal{N}(0, I_d), \qquad \text{and} \qquad X_T \,|\, X_0 = x_0 \sim \mathcal{N}\left(\sqrt{\overline{\alpha}_T}x_0, (1-\overline{\alpha}_T)I_d\right),$$

we can compute

$$\mathsf{KL}\left(p_{Y_T}(\cdot)\,\|\,p_{X_T|X_0}(\cdot\,|\,x_0)\right) = \frac{1}{2}\frac{\overline{\alpha}_T}{1-\overline{\alpha}_T}\left(d + \|x_0\|_2^2\right) + \frac{d}{2}\log(1-\overline{\alpha}_T) \leq \frac{1}{2}\frac{\overline{\alpha}_T}{1-\overline{\alpha}_T}\left(d + \|x_0\|_2^2\right).$$

Using the learning rate schedule in (E.1), we can check that $\overline{\alpha}_T \leq T^{-c_2}$ for some large universal constant $c_2 > 0$; see e.g., Li et al. (2023b, Section 5.1) for the proof. Therefore when $T \geq 2$, we have

$$\mathsf{KL}\left(p_{Y_T}(\cdot)\,\|\,p_{X_T|X_0}(\cdot\,|\,x_0)\right) \leq \frac{d + \|x_0\|_2^2}{4T^{c_2}},$$

which is negligible when $T$ is sufficiently large.

**Optimal solution for** (4.5). It is known that for each $1 \leq t \leq T$, the score function $s_t^\star(\cdot)$ associated with $q_t$ satisfies

$$s_t^\star(\cdot) = \underset{s(\cdot):\mathbb{R}^d \to \mathbb{R}^d}{\arg\min} \, \mathbb{E}_{x \sim q_0, \varepsilon \sim \mathcal{N}(0,I_d)} \left[ \left\| s\left(\sqrt{\overline{\alpha}_t}x + \sqrt{1-\overline{\alpha}_t}\varepsilon\right) + \frac{1}{\sqrt{1-\overline{\alpha}_t}}\varepsilon \right\|_2^2 \right].$$

See e.g., Chen et al. (2022, Appendix A) for the proof. Recall that $\varepsilon_t^\star(\cdot) = \sqrt{1-\overline{\alpha}_t}s_t^\star(\cdot)$, then we have

$$\varepsilon_t^\star(\cdot) = \underset{\varepsilon(\cdot):\mathbb{R}^d \to \mathbb{R}^d}{\arg\min} \, \mathbb{E}_{x \sim q_0, \varepsilon \sim \mathcal{N}(0,I_d)} \left[ \left\| \varepsilon - \varepsilon(\sqrt{\overline{\alpha}_t}x + \sqrt{1-\overline{\alpha}_t}\varepsilon) \right\|_2^2 \right].$$

Therefore the global minimizer for (4.5) is $\widehat{\varepsilon}_t(\cdot) \equiv \varepsilon_t^\star(\cdot)$ for each $1 \leq t \leq T$.

## E.2 TECHNICAL DETAILS IN SECTION 4.2

By checking the optimality condition, we know that $(D_\lambda, G_\lambda)$ is a Nash equilibrium if and only if

$$D_\lambda(x) = \frac{p_{\mathsf{data}}(x)}{p_{\mathsf{data}}(x) + p_{G_\lambda}(x)}, \qquad \text{(optimality condition for } D_\lambda) \tag{E.5}$$

where $p_{G_\lambda} = (G_\lambda)_\# p_{\mathsf{noise}}$, and there exists some constant $c$ such that

$$\begin{cases} -\log D_\lambda(x) + \lambda L(x) = c, & \text{when} \quad x \in \mathsf{supp}(p_{G_\lambda}), \\ -\log D_\lambda(x) + \lambda L(x) \geq c, & \text{otherwise.} \end{cases} \quad \text{(optimality condition for } G_\lambda) \tag{E.6}$$

Taking the approximation $L(x) \approx -\log p_{\mathsf{data}}(x) + C_0^\star$ as exact, we have

$$D_\lambda(x) = \begin{cases} e^{\lambda C_0^\star - c}p_{\mathsf{data}}^{-\lambda}(x), & \text{for } x \in \mathsf{supp}(p_{G_\lambda}), \\ 1, & \text{for } x \notin \mathsf{supp}(p_{G_\lambda}). \end{cases} \tag{E.7}$$

where the first and second cases follow from (E.6) and (E.5) respectively. Then we derive a closed-form expression for $p_{G_\lambda}$.

- For any $x \in \mathsf{supp}(p_{G_\lambda})$, by putting (E.5) and (E.7) together, we have

$$e^{\lambda C_0^\star - c}p_{\mathsf{data}}^{-\lambda}(x) = \frac{p_{\mathsf{data}}(x)}{p_{\mathsf{data}}(x) + p_{G_\lambda}(x)},$$

which further gives

$$p_{G_\lambda}(x) = p_{\mathsf{data}}(x)\left(e^{-\lambda C_0^\star + c}p_{\mathsf{data}}^{\lambda}(x) - 1\right). \tag{E.8}$$

- For any $x \notin \mathsf{supp}(p_{G_\lambda})$, we have

$$-\log D_\lambda(x) + \lambda L(x) \overset{\text{(i)}}{=} \lambda L(x) \overset{\text{(ii)}}{=} -\lambda \log p_{\mathsf{data}}(x) + \lambda C_0^\star \overset{\text{(iii)}}{\geq} c,$$

where step (i) follows from $D_\lambda(x) = 1$, which follows from (E.7); step (ii) holds when we take the approximation $L(x) \approx -\log p_{\mathsf{data}}(x) + C_0^\star$ as exact; and step (iii) follows from (E.6). This immediately gives

$$e^{-\lambda C_0^\star + c}p_{\mathsf{data}}^{\lambda}(x) - 1 = \log\left(-\lambda C_0^\star + c + \lambda \log p_{\mathsf{data}}(x)\right) - 1 \leq 0. \tag{E.9}$$

Taking (E.8) and (E.9) collectively, we can write

$$p_{G_\lambda}(x) = p_{\mathsf{data}}(x)\left(e^{-\lambda C_0^\star + c}p_{\mathsf{data}}^{\lambda}(x) - 1\right)_+. \tag{E.10}$$

On the other hand, we can check that (E.7) and (E.10) satisfies the optimality conditions (E.5) and (E.6), which establishes the desired result.

