# OpenReview forum: "A Score-Based Density Formula, with Applications in Diffusion Generative Models"
_ICLR.cc/2025/Conference — ICLR 2025 Conference Withdrawn Submission_

### Official Review · Reviewer_kQtt · 2024-10-27

**Soundness:** 2
**Presentation:** 1
**Contribution:** 2
**Rating:** 3
**Confidence:** 4

**Summary:**

This work is presented as a theoretical contribution to the field of diffusion models.

The overall structure of the paper mimics the intended objective of this work: to revisit both continuous-time and discrete-time diffusion models to arrive at the (exact and approximate) definition of a density expression for the log data density (in continuous and discrete time), that is used to: i) discuss the validity of an ELBO formulation for the optimization of the parameters of the denoising network of discrete diffusion models, ii) understand the optimization objective in generative adversarial networks, iii) provide a justification for classifier-based guidance in diffusion models, and iv) show that the diffusion loss used in autoregressive models corresponds to an approximate maximum likelihood solution.

**Strengths:**

* It is important to revisit known results that might have been obtained through carefully engineered heuristics, through the lenses of a sound theoretical formalism, such that the community can validate existing choices. The endeavor of this work is in line with this objective, which I think is valuable.
* This work shows that the theory developed to derive an expression for the density of the data distribution can be applied to numerous modeling approaches to generative modeling.
* The mathematical derivations in Appendix A (which are the most important to me), seem correct.

**Weaknesses:**

* Sec. 2: this section repurposes known results from the literature, including [1,2,3], in which it has been shown the equivalence between discrete-time and continuous-time variants of diffusion. Note also that [3], which is not cited by the authors, shows that "*the log-likelihood of score-based models can be tractably computed through a connection to continuous normalizing flows, even though the log-likelihood is not directly optimized by the weighted combination of score matching losses. In particular, it is shown that under a specific weighting scheme the score-matching objective upper bounds the negative log-likelihood, thus enabling approximate maximum likelihood training of score-based models.*"
* In sec 2.1, DDPM are revisited, but mixed with score functions, yielding Eq. 2.3. Why and how does the score function appears in discrete-time diffusion?
* In sec 2.2, I am curious to learn why Eq. 2.4 has been chosen to be so specific, instead of using a more general form with a functional drift term. Here you specify a linear drift whose coefficients explode, compared to the typical variance preserving formulation from [1], as time $t \to 1$.

[1] Song et al. “Score-Based Generative Modeling through Stochastic Differential Equations”, https://arxiv.org/abs/2011.13456

[2] Ho et al. “Denoising Diffusion Probabilistic Models”, https://arxiv.org/abs/2006.11239

[3] Song et al. “Maximum Likelihood Training of Score-Based Diffusion Models”, https://arxiv.org/abs/2101.09258

* Sec. 3: This section displays some calculations that rely on the continuous-time formulation of diffusion processes. Sec. 3.1 begins by focusing on Eq. 2.4, which is the linear variance preserving SDE discussed above. Sec. 3.2 continues the derivations, to relate continuous-time and discrete-time known results, and Sec. 3.3 discusses known results on the equivalence to a probability flow ODE and more recent results on density estimation. What are the main take home messages here? What is the original contribution the authors would like to put forward in this section?
To the best of my understanding, the result in Eq. 3.1.a is an exact formulation for the log likelihood of the data distribution that did not require, as done in [1,3], probability flow ODE equivalence. I followed the proof in Appendix A, and to my eyes it seems correct.
Sec 3.2 should also deserve more insights provided by the authors, as it gives an approximate log density for the discrete case, bypassing the need to work directly in discrete-time. Can we quantify the discretization errors that are introduced by relying on Eq 3.1.c?

* Sec. 4: This is an “application” of the exact log density expression for the data distribution form Sec. 3.
Sec. 4.1 aims at discussing the validity of the ELBO formulation as a good proxy for the log likelihood, to demonstrate that in DDPM optimizing the ELBO is a valid replacement for optimizing log likelihood. This can also be understood from [2] and [1] above, and, for continuous time, is readily discussed in [3], which also shows the similarity (modulo discretization errors and constants) between continuous-time and discrete time formulations. So, what do we learn from the derivations presented in this section that were not directly discussed in these earlier work?
Sec. 4.2 begs for the same question, and should be reviewed in light of an overloaded notation: please check that $z$ is used both as a random variable sampled from a noise distribution, and as a normalizing factor.
Sec 4.3 revisits classifier guidance mechanisms for conditional generation using diffusion models, and offers a critic to some practical heuristics used in recent work, based on the density defined in this paper.
Similarly, Sec. 4.4 revisits autoregressive models in light of the proposed density definition, and suggest that the training objective used in the literature can be viewed as approximate maximum likelihood training.

**Questions:**

* In light of the comments about weaknesses above, can you clearly spell out what are the novel contributions of the submitted article? I am not against revisiting known results to set the stage for the main contributions, but I have the impression that most of the conclusions drawn in Sec. 4, which is where the authors use their revisited formulation of the log data density, have been known to the community, also from the theoretical point of view, and not only from an heuristic perspective. Can you also answer to the questions raised in the "weaknesses" section of this review?

* Despite the intelligible intent of reuniting continuous-time and discrete-time models, I find the exposition of results in Sec. 2 and Sec. 3, according to slightly different formulations than those existing in the literature, is confusing. Is there a way to organize this work such that contributions are more clear, and the implications of the presented theory spelled out well?

* Would you feel comfortable by stating that your exact formulation of the log density of the data distribution as a function of the drift and diffusion terms of the SDEs, or equivalently the log density of the data distribution as a function of the transition kernels and noise of the discrete-time diffusion, as a novel result that has not been discussed in the literature?

---

> ### Author Response · Authors · 2024-12-02
>
> We sincerely thank you for the time and effort you invested in reviewing our submission. We greatly appreciate your feedback and suggestions, which have highlighted several areas for improvement in our work. We have decided to withdraw our submission, which allows us to address the concerns raised and improve the manuscript.

---

### Official Review · Reviewer_dKZW · 2024-11-03

**Soundness:** 3
**Presentation:** 2
**Contribution:** 2
**Rating:** 3
**Confidence:** 3

**Summary:**

This paper derives a density formula for a continuous-time diffusion process, which can be viewed as the continuous-time limit of the forward process of an SGM. The formula relates the target density and the score functions at different time steps. The authors use the formula to show that maximizing the ELBO in DDPM is approximately equivalent to minimizing the KL divergence of the target distribution and the learned distribution.  The authors also apply the approximation to explain the use of score-matching regularization in GAN training, ELBO in diffusion classifier, and diffusion loss is autoregressive models.

**Strengths:**

1. The paper provides a formula relating the target density and the scores at different time steps.

2. The paper shows that maximizing the ELBO in DDPM is approximately equivalent to minimizing the KL divergence of the target distribution and the learned distribution.

**Weaknesses:**

1. The contribution is unclear. The discussion about some important existing results are missing.

2. The applications of the density formula presented here involves approximations, but there is no characterization of the approximation errors.

3. The presentation needs improvement.

**Questions:**

1. Several existing works also explore the relationship between the density and the optimization objectives of diffusion models, e.g. [1] and [2]. What's the relation between the current results and those in existing works.

    [1] Kong et al. "Information Theoretic Diffusion."

    [2] Song et al. "Maximum likelihood training of score-based diffusion models."

2. In general variational inference, for fixed observations, maximizing the ELBO is equivalent to minimizing the KL divergence. What's new in the current results compared to the general observation?

3. Are there error bounds for the various approximations? Without such bounds, why do we expect the interpretation using approximations to be better and more useful than the interpretation using lower bounds?

4. Existing theoretical results have provided error bounds for KL divergence. What's the connection between those results and the current results? What additional insights can the current results bring?

5. What's the advantage of the SDE in (2.4) over the more commonly used O-U process?

---

> ### Author Response · Authors · 2024-12-02
>
> We sincerely thank you for the time and effort you invested in reviewing our submission. We greatly appreciate your feedback and suggestions, which have highlighted several areas for improvement in our work. We have decided to withdraw our submission, which allows us to address the concerns raised and improve the manuscript.

---

### Official Review · Reviewer_SVbJ · 2024-11-04

**Soundness:** 4
**Presentation:** 2
**Contribution:** 2
**Rating:** 3
**Confidence:** 3

**Summary:**

Despite empirical advances, the theoretical foundation for why optimizing the evidence lower bound (ELBO) on the log-likelihood is effective for training diffusion generative models, such as DDPMs, remains largely unexplored.  The authors proposed to address this question by establishing a density formula for a continuous-time diffusion process, which can be viewed as the continuous-time
limit of the forward process in an SGM. The formula shows that the variational gap is negligible.

**Strengths:**

The analysis of the variational gap for the continuous-time approximation of the discrete alternative of diffusion models, such as DDPM, is missing. The authors conducted clear and solid derivations to show why the variational gap is negligible, which provides proof for the empirical usage that the ELBO matches the true objectives.

**Weaknesses:**

1. despite the soundness of the derivations, I found that the goal of this paper is not very interesting.

2. insights on GANs are not clear to me (a layperson in GAN).

**Questions:**

your SDE $d X_t = - \frac{1}{2(1-t)} X_t d t + \frac{1}{\sqrt{1-t}} d B_t $  is a special case in Song's SDE in [1], which follows that

$$d X_t = -\frac{1}{2} \beta_t X_t dt + \sqrt{\beta_t} d B_t$$

It appears that there are countless choices of $\beta_t$. Why do you claim $\beta_t = \frac{1}{1-t}$ is the continuous-time limit of the aforementioned forward process in section 2.1 and is more preferred than Song's linear version $\beta_t = \beta_{\min} + t (\beta_{\max} - \beta_{\min})$?

[1] Score-Based Generative Modeling through Stochastic Differential Equations. ICLR'21.

---

> ### Author Response · Authors · 2024-12-02
>
> We sincerely thank you for the time and effort you invested in reviewing our submission. We greatly appreciate your feedback and suggestions, which have highlighted several areas for improvement in our work. We have decided to withdraw our submission, which allows us to address the concerns raised and improve the manuscript.

---

### Official Review · Reviewer_WunF · 2024-11-04

**Soundness:** 2
**Presentation:** 2
**Contribution:** 1
**Rating:** 3
**Confidence:** 4

**Summary:**

This paper considers a density formula based on score estimation to analyze Denoising Diffusion Probabilistic Models (DDPMs). Using this formula, the authors provide a theoretical basis for why optimizing the evidence lower bound (ELBO) serves as an effective approach for training these models. The paper addresses the problem of the understanding of ELBO optimization for diffusion models, adding theoretical context to a widely used empirical technique.

The analysis extends to practical implications across different generative modeling contexts, including applications to GAN regularization, diffusion classifiers, and autoregressive models. By investigating these areas, the authors demonstrate how insights from the density formula can support training and optimization practices in various generative frameworks. This broad applicability suggests that the theoretical findings may be interesting to both foundational research and practical applications in generative modeling.

**Strengths:**

The paper focuses on the theoretical understanding of score-based generative models (SGMs) by applying a density formula to explain why optimizing the evidence lower bound (ELBO) effectively supports training for diffusion models like DDPMs. By investigating the theoretical aspects  behind ELBO optimization, the authors promote a more rigorous basis for diffusion models.

Additionally, the paper extends the implications of this analysis to areas such as GAN regularization, diffusion classifiers, and autoregressive models, illustrating the potential for these findings to enhance model training practices across various generative frameworks

**Weaknesses:**

The biggest weakness of the work lies in the lack of novelty and the positioning with respect to the literature.
In particular, it is not evident how the work differs from known results such as [1], where Thm 4+ eq (25) and the comment in eq(29) seems to provide a result which is even more general than the one discussed by the authors.  It is worth mentioning that related results which the authors do not cite in their work are also presented in [2], in particular Thm 3.


One other big limitation is that the authors derive their connection between continuous and discrete time (with an **approximated score**) by a sequence of approximations, without discussing properly the impact of these. A clear quantification analysis would greatly strengthen the paper.



[1] Huang et al., A Variational Perspective on Diffusion-Based Generative Models and Score Matching, (NeurIPS 2021)

[2] Song et al., Maximum Likelihood Training of Score-Based Diffusion Models (NeurIPS 2021)

**Questions:**

Please refer to weaknesses  section. Also, why do the authors consider the particular SDE in lines 76-77 and not a more generic one?

---

> ### Author Response · Authors · 2024-12-02
>
> We sincerely thank you for the time and effort you invested in reviewing our submission. We greatly appreciate your feedback and suggestions, which have highlighted several areas for improvement in our work. We have decided to withdraw our submission, which allows us to address the concerns raised and improve the manuscript.

---

### Note · Authors · 2024-12-02

I have read and agree with the venue's withdrawal policy on behalf of myself and my co-authors.